# Global analysis of contact-dependent human-to-mouse intercellular mRNA and lncRNA transfer in cell culture

**Sandipan Dasgupta**[1], **Daniella Y Dayagi**[1], **Gal Haimovich**[1]*, **Emanuel Wyler**[2], **Tsviya Olender**[1], **Robert H Singer**[3], **Markus Landthaler**[2], **Jeffrey E Gerst**[1]*

[1]Department of Molecular Genetics, Weizmann Institute of Science, Rehovot, Israel; [2]Berlin Institute of Medical Systems Biology and Systems Biology, Max Delbruck Center for Molecular Medicine, Berlin, Germany; [3]Department of Anatomy & Structural Biology, Albert Einstein College of Medicine, New York, United States

**Abstract** Full-length mRNAs transfer between adjacent mammalian cells via direct cell-to-cell connections called tunneling nanotubes (TNTs). However, the extent of mRNA transfer at the transcriptome-wide level (the 'transferome') is unknown. Here, we analyzed the transferome in an in vitro human-mouse cell co-culture model using RNA-sequencing. We found that mRNA transfer is non-selective, prevalent across the human transcriptome, and that the amount of transfer to mouse embryonic fibroblasts (MEFs) strongly correlates with the endogenous level of gene expression in donor human breast cancer cells. Typically,<1% of endogenous mRNAs undergo transfer. Non-selective, expression-dependent RNA transfer was further validated using synthetic reporters. RNA transfer appears contact-dependent via TNTs, as exemplified for several mRNAs. Notably, significant differential changes in the native MEF transcriptome were observed in response to co-culture, including the upregulation of multiple cancer and cancer-associated fibroblast-related genes and pathways. Together, these results lead us to suggest that TNT-mediated RNA transfer could be a phenomenon of physiological importance under both normal and pathogenic conditions.

*For correspondence:
gal.haimovich@weizmann.ac.il (GH);
jeffrey.gerst@weizmann.ac.il (JEG)

## Editor's evaluation

This study presents an important finding on the characterization of cell contact-dependent transfer of mRNAs between human MCF7 breast cancer cell line, and immortalized mouse embryo fibroblasts (MEFs) grown in co-culture. The evidence supporting the conclusions is compelling, with rigorous data analysis and multiple approaches to address the specific questions of the sequences of the transferred mRNAs, the presence of specific sequences targeting this transfer, the cell-contact-dependent mechanisms, and the transcriptional consequences of the transfer. This work will be of interest to cell biologists and biologists.

## Introduction

RNA molecules can act as mediators of intercellular communication in plants and animals during normal growth and development, as well as different pathologies such as viral infections or cancer (*Dreux et al., 2012*; *Haimovich et al., 2021*; *Lu et al., 2019*; *O'Brien et al., 2020*; *Ramachandran and Palanisamy, 2012*; *Valadi et al., 2007*). Although initial evidence of the intercellular transfer of RNA was found in the early 1970s (*Kolodny, 1971*; *Kolodny, 1972*), the mechanism of transfer gained attention much later. The prevalent mechanism by which RNAs are thought to be transferred is through extracellular vesicles (EVs), which include apoptotic bodies, microvesicles and exosomes.

Best studied are exosomes, which are vesicles of endocytic origin with sizes ranging from 40 to 100 nm and can transfer information in the form of proteins, lipids, DNA and RNA (*O'Brien et al., 2020*). Released by numerous cell types (e.g. immune cells, neuronal and cancer cells *Mittelbrunn et al., 2011*; *Villarroya-Beltri et al., 2013*), they are secreted into extracellular media and body fluids (e.g. saliva, blood plasma, breast milk, and urine) (*Zhang et al., 2019b*). Profiling of their RNA content using DNA microarrays or RNA sequencing has revealed that while multiple species of RNAs are present in exosomes (e.g. snoRNA, siRNA, Y-RNA, lncRNA, and vault RNA), work has mainly focused on microRNAs (miRs; *Li et al., 2014*; *Valadi et al., 2007*). Exosomes have also been suggested to transfer mRNA (or at least fragments of mRNA). One report found that GFP mRNA in exosomes was taken up by a colon cancer cell line and translated (*Jiang et al., 2015*). However, exosomes may not be the sole agent by which RNAs transfer from one cell to another. In fact, multiple studies show that only small sized RNAs (e.g. miRs) and fragments of mRNA are likely to be enriched in the exosomes (*Batagov and Kurochkin, 2013*; *Pérez-Boza et al., 2018*; *Zhang et al., 2018*). Thus, exosomes may not be the preferred mode of full-length mRNA transport between cells. Interestingly, ARC and PEG10 mRNAs encode proteins that can form retroviral-like capsids and package their own mRNAs for transfer to neighboring cells (*Ashley et al., 2018*; *Pastuzyn et al., 2018*; *Segel et al., 2021*). However, this form of transfer is probably limited to a small set of mRNAs encoding retroviral-like capsid proteins.

Our lab has shown that full-length messenger RNA (mRNA) molecules can transfer between mammalian cells, but do so via tunneling nanotubes (TNTs) (*Haimovich et al., 2017*; *Haimovich and Gerst, 2019*). TNTs are cytoplasmic connections extending between cells that are thinner and longer than other dynamic cellular protrusions, such as lamellipodia and filopodia (*Cordero Cervantes and Zurzolo, 2021*). TNTs have been shown to differ significantly from filopodia in both actin architecture and overall structure (*Korenkova et al., 2020*; *Ljubojevic et al., 2021*; *Sartori-Rupp et al., 2019*). TNTs, which are typically 0.2–1 µm in diameter and up to >150 µm long, were shown to transfer organelles (*Goodman et al., 2019*; *Kolba et al., 2019*; *Murray and Krasnodembskaya, 2019*; *Wang et al., 2011*; *Zou et al., 2020*), bacteria (*Kim et al., 2019*; *Onfelt et al., 2006*), viruses (*Eugenin et al., 2009*; *Guo et al., 2016*; *Panasiuk et al., 2018*; *Roberts et al., 2015*; *Tiwari et al., 2021*), proteins (*Biran et al., 2015*), and microRNAs (*Lu et al., 2019*; *Valadi et al., 2007*). TNTs have been reported to be involved in a multitude of biological processes, such as stem cell differentiation, immune response, and neurodegenerative diseases, among others (*Abounit et al., 2016*; *Reichert et al., 2016*; *Zhu et al., 2021*). TNT-mediated RNA transfer has been demonstrated in only a few physiological contexts (reviewed in *Haimovich et al., 2021*). In addition to our work (*Haimovich et al., 2017*; *Haimovich and Gerst, 2019*), published RNA-sequencing data and qPCR analysis were used to detect exosome-independent transfer of keratinocyte-specific mRNAs to Langerhans cells (*Su and Igyártó, 2019*), and GFP and GNAT1 mRNAs were shown to transfer in vivo in a photoreceptor transplantation model (*Ortin-Martinez et al., 2021*).

Previously, we used a simple 2D co-culture model of donor and acceptor cells along with single molecule fluorescent in situ hybridization (smFISH) to quantitatively study TNT-mediated mRNA transfer (*Haimovich et al., 2017*; *Haimovich and Gerst, 2019*). Using this approach, we found that several full-length endogenous or ectopically expressed mRNAs undergo intercellular transfer between a variety of cell types, including immortalized cells, primary cells, and even human-mouse cell co-cultures. While the level of transfer was often <1% of the endogenously expressed message in donor cells, the most abundantly transferred mRNA was that of β-actin mRNA, which transferred at a level of up to 5% in some experiments. Initial analyses showed that transfer is influenced by the identity of the donor cell, by the expression level of the RNA in donor cells and by stress conditions. We further found that the transfer of mRNAs is inhibited by both cytoskeletal and small GTPase inhibitors, as well as by the binding of multiple MS2 coat proteins (MCPs) to a 24xMS2 stem-loop aptamer sequence in the mRNA (*Haimovich et al., 2017*). Inhibition by the MS2 system could be explained by extensive MCP binding and hindrance to transfer or by interference with a yet-to-be-identified protein that may coat the transferred mRNA (*Haimovich and Gerst, 2019*).

Here, we used human MCF7 breast cancer cell-line and immortalized mouse embryo fibroblast (MEF) cells grown in co-culture to define the extent of the RNA transferome and the effect of co-culture upon the native transcriptome in vitro. By employing deep sequencing, we found that nearly all mRNAs transfer and in a manner that strongly correlates with the donor cell expression level.

Moreover, by increasing gene expression using reporter mRNAs we could verify that increased expression results in increased transfer. Similar to our previous findings (*Haimovich et al., 2017*), the overall level of transfer was found to be <1% of the expression level in the donor cells and was dependent upon cell-cell contact. To explore the mechanism of mRNA transfer, we used two approaches aimed at identifying *cis* RNA elements that might be required. First, we searched for previously identified EV-targeting motifs or novel unique elements present in transferred RNAs using bioinformatics and second, we fused short segments of β-actin mRNA to a reporter mRNA to test if they were able to increase transfer. However, neither approach revealed the presence of an element that enhances RNA transfer, indicating that the process is inherently non-selective. We used imaging and a Transwell-like system (*i.e.* quadrapod), to show that RNA transfer is likely to be contact-dependent via TNTs, as shown for a few example mRNAs. Lastly, we detected significant differential changes in the native transcriptome of the MEF cells in response to co-culture with MCF7 cells, including the upregulation of multiple cancer- and cancer-associated fibroblast (CAF)-related genes and pathways. Together, these results lead us to suggest TNT-mediated RNA transfer could be a phenomenon of physiological importance under both normal and pathogenic conditions.

## Results

### An antigen-based cell sorting method to separate human and mouse cells after co-culture

In order to study genome-wide RNA transfer, it is essential to separate the two cell types after co-culture and prior to RNA extraction and downstream analysis (schematic shown in *Figure 1A*). We previously demonstrated using smFISH that up to 2% of β-actin mRNA (*e.g.* ~30 copies/cell) can undergo transfer from MEFs to human MCF7 cells within 12 hr of co-culture (*Dasgupta and Gerst, 2020*). As in that study, here we used mouse MEFs tagged with 24 repeats of the MS2 coat protein (MCP)-binding sequence (MBS) between the ORF and 3′ UTR of both endogenous alleles of β-actin (referred to hereafter as 'MBS-MEFs'; *Haimovich and Gerst, 2019*; *Lionnet et al., 2011*) and human MCF7 cells. MCF7 cells specifically express a cell surface molecule, CD326 or Epithelial Cell Adhesion Molecule (EpCAM). Thus, we employed magnetic beads conjugated to anti-CD326 antibodies in order to separate heterologous cell populations of MBS-MEF and MCF7 cells to a high degree of purity. It is critical to have complete cell separation (~100%), since even a small contamination (e.g. 0.1–0.3%) of donor cells in the isolated acceptor cells can lead to a high background relative to the transferred RNA signal, particularly when the level of RNA transfer is expected to be low. To eliminate background signals originating from incomplete cell separation, we compared our co-culture samples to MEFs and MCF7 single cultures that were grown separately, harvested, mixed after cell harvest, and immediately separated (referred to hereafter as the 'Mix'). Singlecell cultures were used as controls to determine endogenous gene expression levels before co-culture, as well as to eliminate background signals originating from downstream RNA sequencing procedures. Once separated, we checked the efficiency of cell sorting from the Co-culture, Mix, and Single cell culture samples by flow cytometry. Based on the purity of human and mouse cell fractions, as determined by flow cytometry (*Figure 1B* and *Figure 1—figure supplement 1A*), we found that the mouse MEF-enriched fractions were much better sorted, as compared to the human MCF7 cell-enriched fractions (*Supplementary file 1 -table 1*). We found that the MEF population was essentially free from MCF7 cells in both the Co-culture (0%) and Mix-derived populations (0.025% on average), whereas the MCF7 fractions contained an average of 0.035% MEFs from the Co-culture samples and an average of 0.22% MEFs from the Mix samples, after sorting.

Next, total RNA from the sorted cell populations was sequenced to identify the transferred RNAs, as well as to measure the native transcriptomes from single cell cultures. We first considered how to treat the raw reads with respect to the length and type of fragment (e.g. single or paired-ends) to obtain the best results (*Figure 1—figure supplement 1B*). Due to regions with a high degree of homology between the human and mouse transcriptomes, using sub-optimal length reads may lead to the mapping of human reads to the mouse genome and *vice versa*. Based on a previous in silico simulation of mapping 25, 50, 75, and 100 bp long single- and paired-end reads from the human transcriptome to the mouse transcriptome, we observed that short (25 bp), single-end reads have a very high (>85%) non-specific alignment with the mouse genome, which was drastically reduced by using

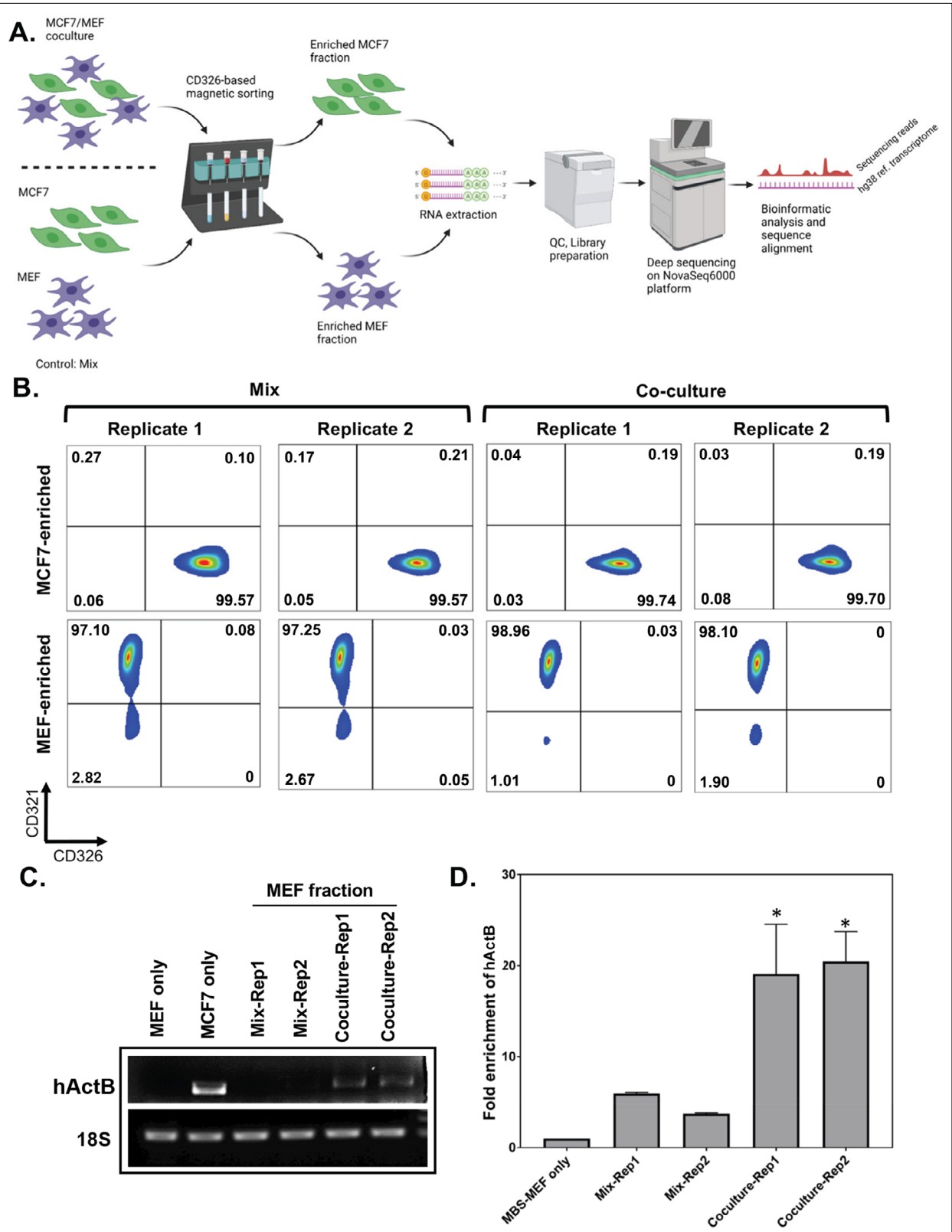

**Figure 1.** Affinity purification of single populations of MCF7 and MEF cells following mix/co-culture. (**A**) Schematic representation of the magnetic bead-based sorting and RNA sequencing to identify mRNAs transferred between two cell types. Between 2 and 3 million each of human and mouse cells (i.e. MCF7 cells and MBS-MEFs, respectively) are either co-cultured or only mixed before cell sorting. The heterologous cell population is then sorted using magnetic microbeads conjugated to an anti-CD326 antibody specific to MCF7 cells. The sorted cells are then analyzed by RNA-sequencing

*Figure 1 continued on next page*

*Figure 1 continued*

to identify the transferred RNAs. (**B**) Flow cytometry profiles of the sorted cell populations of MCF7 and MBS-MEF cells following affinity purification. Post-sorting, cell suspensions were counterstained with a human CD326-Alexa 488 antibody that labels only the MCF7 cells and a mouse CD321-PE antibody that labels the MBS-MEF cells. The sorted cell populations were analyzed by flow cytometry using the Alexa Fluor-488 and R-PE windows. (**C–D**) RT-PCR (**C**) and RT-qPCR (**D**) validation of transfer of β-actin mRNA from MCF7 to MBS-MEF cells in co-culture. Total RNA from one replicate of a MBS-MEF and MCF7 single culture and two replicates of MBS-MEF fractions from Mix and Co-culture were collected for analysis by semi-quantitative RT-PCR (25 cycles) (**C**) (representative image from three replicas) and RT-qPCR (**D**) using primers specific for human β-actin. 18 S rRNA was amplified as an internal control. The graph in **D** shows the average of three repeats. * - p≤0.05. See *Figure 1—source data 1* for complete gel images of panel **C**.

The online version of this article includes the following source data and figure supplement(s) for figure 1:

**Source data 1.** Full agarose gel images of the RT-PCR depicted in *Figure 1C*.

**Figure supplement 1.** Flow cytometry profiles of single cultures of MBS-MEF and MCF7 cells and bioinformatic pipeline to identify species-specific mRNAs.

longer (>100 bp) and paired-end reads (*Dasgupta and Gerst, 2020*). We also noted that using suboptimal sequencing depth (i.e. <25 million reads per sample) can lead to missing rare transcripts (data not shown). Hence, we chose to sequence the samples to obtain maximal depth (e.g. >100 million reads per sample) and using 2x150 base-pair paired-end reads. We obtained between 94–281 million reads per sample at an average of 168 million paired-end reads per sample (*Source data 1 - table 1*). Between 90–92% of the reads were uniquely aligned to the respective reference genomes, which is an indicative of the high quality of library preparation and it was either at par or better than typical unique alignment rates (*Mortazavi et al., 2008*; *Sarantopoulou et al., 2019*; *Sun et al., 2013*; *Zhao et al., 2015*). More than 14,000 mouse and 16,000 human genes were annotated to their respective RefSeq database. Such high depths of sequencing ensure that low levels of mRNA transfer could be detected.

Initial analysis of mouse-derived RNA in the human transcriptome revealed that more mouse RNA was present in MCF7 cells obtained from the Mix than from MCF7 cells in Co-culture (the percentage of unique mouse-aligned reads in the Mix samples was actually greater than that of the Co-culture; *Source data 1 - table 1*). This result is probably due to the high level of MBS-MEF contamination (e.g. 0.17–0.27%) in the Mix samples (*Figure 1B* and *Supplementary file 1 - table 1*). In contrast, the level of human MCF7 cell contamination in the MEF fraction was very low in the Mix (≤0.05%; e.g. 0% in one replica and 0.05% in the second replica) and essentially absent in the Co-culture samples (e.g. 0% in both replicas). In addition, we found small sub-populations of double-stained and unstained cells within the purified populations that we suspect are mostly MEFs (see Materials and methods). These sub-populations were greater in the replicas of Mix-derived MEFs vs. the Co-culture-derived MEFs (i.e. 0.08% and 0.03% double-stained, and 2.8% and 2.67% unstained in Mix samples vs. 0% and 0.03% double-stained, and 1% and 1.9% unstained in the Co-culture samples). As a consequence, if these double-stained and unstained cells had contributed to the background of human reads in the MEFs, we would have expected to have many more human reads in the Mix-derived MEFs. However, the percentage of unique human-aligned reads in Co-culture samples (versus the Mix samples) was substantially greater (e.g.~1.55% [1.71% and 1.39%] versus 1.13% [1.11% and 1.15%]), respectively (*Source data 1 - table 1*). This difference becomes more apparent after subtraction of the background (arising from library preparation and RNA sequencing), which is reflected in the single culture MEF-enriched samples and constituted 1.055% (e.g. 1.04 and 1.07% for both replicas). Thus, the true proportion of unique human reads in the Mix versus the Co-culture samples was 0.075% and 0.495%, respectively, giving a read ratio that was 6.6-fold higher for MEFs in Co-culture versus MEFs in the Mix. The background in the single cell mouse cultures likely arises from index hopping, due to the use of single-index adaptors in library preparation and next generation sequencing, which leads to read misalignment.

Given the high degree of purity of the MEF cell population after co-culture and sorting, we focused only on human RNAs present in the mouse-derived transcriptome samples (i.e. human-to-mouse RNA transfer). In total, 10,566 transcripts of human genes were detected in Co-culture-derived or Mix-derived mouse-enriched samples (*Figure 2A*; *Source data 1 - table 2*). Of those, 7,504 genes had RPM counts of more than 10 in both co-culture replicates (*Source data 1 - table 2*), of which 7501 genes had a fold-change (FC) of >1 (*i.e.* more reads in the Co-culture vs Mix), and 6827 genes had FC of ≥2 (*Source data 1 - table 2*). As a positive control for the experiment, we first confirmed the

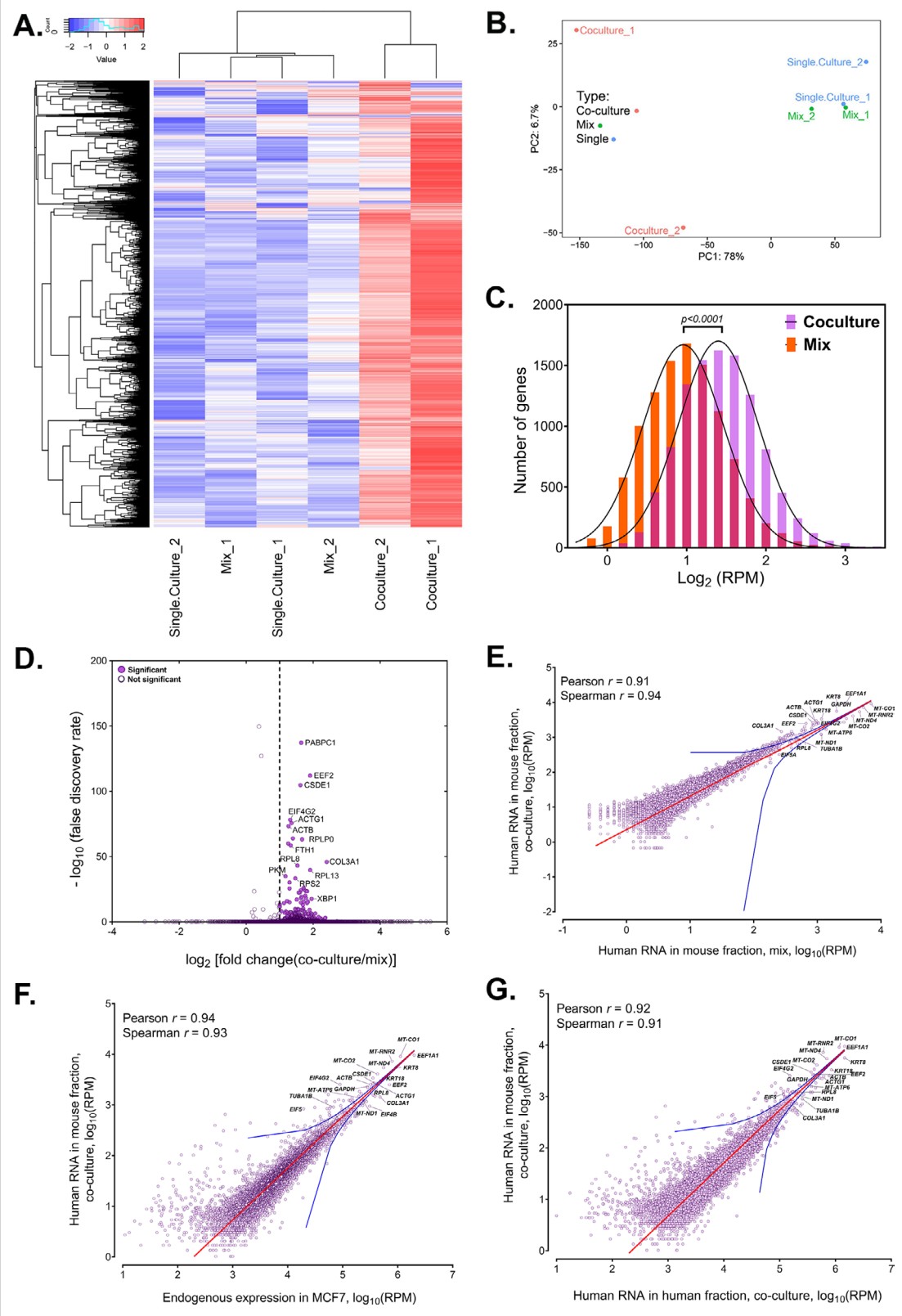

**Figure 2.** Analysis of human RNAs present in mouse-enriched fractions and identification of transferred mRNAs. (**A**) Heat map analysis showing the transferred human RNA in mouse samples. RPM counts of human-specific genes across single culture, mix and co-culture samples of mouse-enriched fractions were Z-normalized across rows and plotted using the G-Plot package of R. Each row indicates a single gene and each column represents a sample. Cluster analysis of the samples was done by 'maximum' distance function. (**B**) PCA analysis of the different samples. Human-specific reads from

*Figure 2 continued on next page*

*Figure 2 continued*

mouse samples were analyzed and plotted, as a Principal Component Analysis graph using the base functions of R. The scales represent the numbers obtained after PCA transformation and represent the percent variability in the samples contributed by each principle component (78% for PC1 and 6.7% for PC2). Red icon: Co-culture, Blue icon: Single Culture and Green icon: Mix. (**C**) Read distribution between the Co-culture and Mix samples. The lognormal distribution of the reads in Co-culture (purple bars) and Mix (orange bars) were plotted and fitted to a Gaussian curve, and revealing that mRNAs from almost the entire human trancriptome underwent transfer. This was depicted by a shift of the "Co-culture" curve to the right. The red bars represents the overlap between the two distributions. The p value for the average of all human genes in the Gaussian is given. (**D**) Identification of RNAs that underwent robust transfer. A dot plot representing the results from a single-tailed, unpaired *t*-test between the two replicates of Co-culture and two replicates of Mix was performed to identify the genes exhibiting the highest transfer. Significance threshold (dashed line): Fold change ≥2; false discovery rate (FDR)<0.05. Closed circles: Significant genes, open circles: Statistically insignificant genes. An FDR of 0.05 roughly implies that 5% of significant tests will result in false positives. (**E–G**) Identification of human RNAs in mouse cells relative to gene expression levels. Linear regression analyses of human-specific read counts from the mouse fraction of the Co-culture samples with respect to the human reads from mouse fraction of Mix samples (**E**) with the reads from the MCF7 Single culture (**F**), or with reads from MCF7 in co-culture (**G**), which provide various ways to look at the level of endogenous expression. The red line indicates the linear regression line, while the blue lines indicate the upper and lower bounds of the 99% confidence internal. Pearson and Spearman coefficients are indicated.

The online version of this article includes the following source data and figure supplement(s) for figure 2:

**Figure supplement 1.** Analysis of the fold-change (FC) of human RNA transferred to mouse cells.

**Figure supplement 2.** GO annotation and pathway analysis of the highly transferred genes.

**Figure supplement 2—source data 1.** This file contains GO terms analysis of transferred mRNAs.

transfer of human β-actin mRNA by both RT-PCR and qRT-PCR of the mouse-derived RNA using human β-actin-specific primers (*Figure 1C and D*, respectively). Both detection methods revealed a significant enrichment of human β-actin mRNA in the MBS-MEFs after co-culture.

## The human transferome encompasses the major fraction of the transcriptome

Whole transcriptome profiling of the human- and mouse-enriched samples from Co-culture and control (Mix) samples revealed the robust presence of human-specific reads in the mouse samples (*Figure 2A*). Principal Component Analysis of these samples indicated that Co-culture samples have a similar profile of human-aligned genes and were different from control Mix and Single culture samples, as expected (*Figure 2B*). Note that the difference between the two Co-culture samples on the Y-axis shows a small variation of 6.7%. This difference likely results from small differences in the individual Co-culture samples (as such differences are often observed within replicas of RNA-seq experiments) and not via large differences in the measured transferomes. This indicates that the Co-culture samples were quite similar overall. The number of cross-species reads is expected to be higher in Co-culture samples (due to the transfer of RNAs) than Mix samples (assuming the number of residual cells after sorting is similar). Comparison of the results from two biological replicates of Co-culture, Mix and Single culture samples revealed that most transcripts of human genes are able to transfer to varying degrees (*Figure 2A*; *Source data 1 - table 2*).

As RNAs from most human genes are transferred, the distribution of these RNAs in the mouse cells is globally shifted to the right, *i.e.* the average number of reads per human gene in mouse samples is much higher in Co-culture samples as compared to the Mix samples (*P*=0.0001 for all genes; *Figure 2C*). By employing a single-tailed unpaired *t*-test, we found that 283 mRNAs robustly transferred from human (MCF7) cells to mouse (MBS-MEF) cells with a fold-change of more than 2 (*Figure 2D*, *Source data 1 - table 3*). Many of the identified mRNAs included cytoskeletal components (ACTB, ACTG), translation factors (EEF1A1, EEF2), ribosomal proteins (RPL8, RPL4, RPL7) and other 'housekeeping' genes (GAPDH, PABPC1). Among this group we also identified mRNAs encoding multiple types of keratins (e.g. KRT8, 18, 19, and 80), similar to those reported to transfer from keratinocytes to epidermal-resident Langerhans cells (*Su and Igyártó, 2019*). It should be noted that this statistical test is underpowered due to the low number of replicates and, hence, likely underestimates the number of mRNAs undergoing robust transfer. Essentially, these 283 genes represent the most abundantly transferred mRNAs in terms of absolute number. We note that amongst the 7 endogenously expressed mRNAs previously shown by us to transfer (*Haimovich et al., 2017*), only ACTB and CCND1 underwent robust transfer. This may also be due to the lower endogenous expression levels of these genes (e.g. BRCA1 [3252 RPM], HER2/ERBB2 [25,690 RPM], MITF [436 RPM], MT2A

[4383 RPM] and SERP2 [undetected]) in MCF7 cells in comparison to CCND1 (108670 RPM) or ACTB (320214 RPM) (*Source data 1 - table 4*). Indeed, BRCA1 and ERBB2 gene expression levels in MCF7 cells are respectively 1.5-fold and ~45-fold lower than in the HEK293 and SKBR3 cells (Human Protein Atlas; *Uhlén et al., 2015*) in which we already observed low levels of transfer (*Haimovich et al., 2017*). This further implies that low gene expression can affect the detection of transfer of these mRNAs by RNA-seq.

Since the 283 abundantly transferred mRNAs were endogenously expressed at high levels in MCF7 cells, we wondered if transfer is dependent upon the level of expression. To determine if the level of mRNA transfer correlates with endogenous RNA expression in donor cells, we performed linear regression analysis of the human reads found in Co-culture samples with those of the Mix (*Figure 2E*) and in human Single cultures (*Figure 2F*). Human reads from the MEF-enriched fraction of the 'Mix' sample can be attributed to the residual presence of MCF7 cells after magnetic bead sorting, whereas those in MCF7 single cultures are representative of the initial mRNA expression levels prior to co-culture. We found that the number of transferred human RNAs strongly correlated with their expression levels (Pearson and Spearman coefficients >0.9; *Figure 2F*), most falling within the 99% confidence interval of the regression. This indicates that the level of gene expression prior to co-culture can be a dominant predictor of RNA transfer. A similar correlation is also seen between the level of transferred human RNAs and their endogenous expression in MCF7 cells after co-culture (*Figure 2G*). This is in agreement with our previous observation, which showed that the transfer of MS2-tagged human Cyclin D1 (CCND1-MBS) mRNA from HEK293 cells to MEFs is enhanced upon increased gene expression in the donor cells (i.e. using a stronger promoter). Likewise, a strong correlation was observed between ACTB-MBS expression levels and ACTB-MBS mRNA transfer when comparing between immortalized and primary donor MBS-MEFs (*Haimovich et al., 2017*).

Next, we looked at the relationship between the endogenous expression in the donor MCF7 cells and the fold-change (FC) of transfer (i.e. the fold increase of RPM counts in Coculture samples divided by the Mix samples). When looking at the whole transcriptome, we found no correlation between endogenous RNA expression in donor cells to the $\log_2$FC of human RNAs in co-cultured cells (*Figure 2—figure supplement 1*, Pearson coefficient = 0.07). We noted that unlike the 283 robustly expressed and transferred RNAs described above, mitochondria-encoded RNAs (e.g. 13 mRNAs and 2 rRNAs) showed poor transfer ($\log_2$FC = 0.03–0.45) despite their high expression. This result further strengthens our confidence that we measure actual RNA transfer and not contamination. Most of the genes with $\log_2$FC <0 (i.e. presumably RNAs that do not transfer) and $\log_2$FC >3 (i.e. presumably highly transferred RNAs) have values of less than 10 RPM (*Source data 1 - table 4*). Thus, we cannot determine if these $\log_2$FC values are accurate or artifacts relating to the low number of reads.

We further observed that the median percentage of transfer for all human RNAs was low [0.34%, $\log_{10}$ (percentage of transfer)=–0.44] (*Figure 3A*, *Source data 1 - table 4*). Although the absolute number of transferred RNAs (using RPM as a measure) depended on gene expression level in the donor cells, the percentage of RNA transfer did not correlate linearly with gene expression (*Figure 3B*, *Source data 1 - table 4*). This result indicates that only a low proportion of RNAs from the mammalian transcriptome is likely to undergo transfer to neighboring cells. Next, we examined whether the percentage transfer of mRNA correlates with mRNA stability. To check this, we compared the percent transfer data with a recently published genome-wide analysis of half-lives of mRNAs in K562 cells (*Blumberg et al., 2021*). In total, 4972 genes were correctly annotated between the two datasets. We found that the percentage of transfer of the total RNA population, as well as that of the group of 283 robustly transferred mRNAs, was largely independent of mRNA stability (*Figure 3C*, *Source data 1 - table 5*). Interestingly, we could also detect the transfer of mRNAs from 12 primate-specific genes (*e.g.* DHRS4L2, GTF2H2C, NBPF10, NBPF14, ALG1L, CBWD2, APOL2, ZNF43, ZNF726, ZNF816, ZNF680, and ZNF718) in co-cultured MEFs. These genes are highly expressed in cortical regions of primate brains and are not reported to have any mouse orthologues (*Florio et al., 2018*). The endogenous expression of most of these genes in MCF7 cells was found to be low, as was the level of transfer (*Source data 1 - table 6*).

We also examined the RNA-seq data for the presence of long non-coding RNAs (lncRNAs). We detected the expression of 174 lncRNAs in MCF7 Single cultures (out of 5301 known lncRNAs) and identified 102 having >10 RPM in Coculture samples - 100 of which were transferred with a FC >2 (*Source data 1 - table 7*). We noted that 5 lncRNAs were among the most highly endogenously

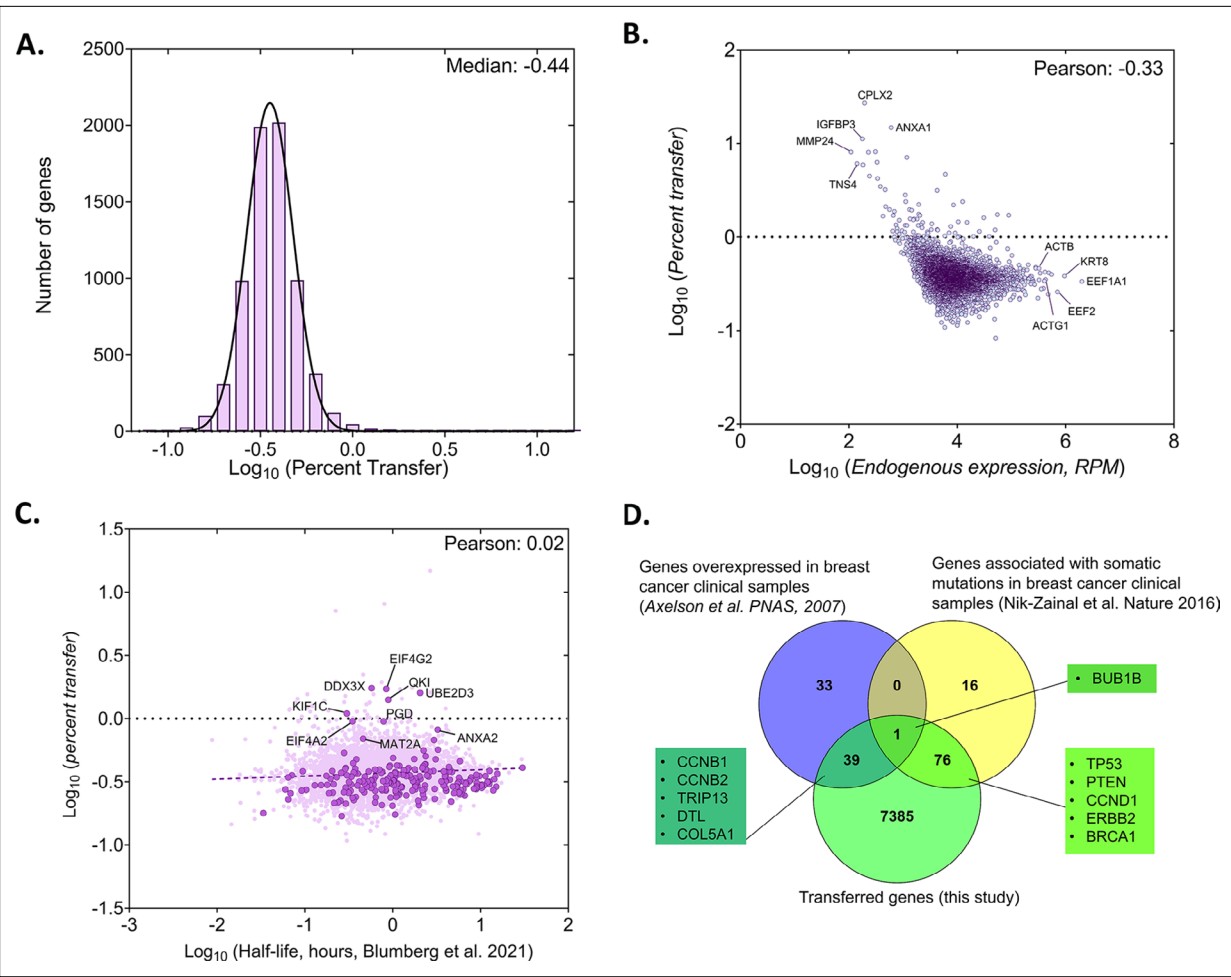

**Figure 3.** Analysis of the percentage of human RNA transferred to mouse cells. (**A**) Genome wide distribution of proportion of donor mRNAs that undergo transfer. The black curve indicates the best-fit Gaussian distribution. Log of median percent transfer is indicated. (**B**) Linear regression analysis of the percentage transfer for each gene with the donor endogenous level of expression. Each circle represents an individual. Pearson correlation is indicated. (**C**) Linear regression analysis of the percentage transfer of each mRNA with its half-life. Stability data of each mRNA was derived from a recent analysis of genome-wide mRNA half-lives (**Blumberg et al., 2021**). RPM count of transferred mRNAs (y-axis) and half-lives of mRNAs (x-axis) of 4972 annotated genes are plotted. Solid dark purple dots indicate 223 of the 283 robustly transferred mRNAs identified in **Figure 2D** and **Source data 1 - table 3**. The purple solid line indicates the linear regression line; Pearson coefficient is indicated. (**D**) Breast cancer-specific overexpressed and mutated genes were found to undergo mRNA transfer from MCF7 cells to MEFs. This list of human-mapped genes in mouse enriched fraction from co-culture sample was compared to a set of 73 genes that were found to be overexpressed in 184 breast cancer samples from 11 patients, compared to 8 samples from healthy tissues (**Axelsen et al., 2007**) and to a set of 93 genes that were found to be frequently mutated in 560 breast cancer samples (**Nik-Zainal et al., 2016**). A representative list of genes from the indicated overlaps are mentioned in the boxes. Only the genes with RPM counts of >10 in both replicates of co-culture and FC >1 were considered for this analysis (7500 genes).

expressed genes in MCF7 cells and amongst the 283 robustly transferred RNAs (**Source data 1 - table 3**).

Since the donor human cell line (MCF7) was derived from breast cancer tissue, we wondered if the mRNAs of human genes known to drive cancer undergo transfer to neighboring acceptor fibroblasts (MBS-MEFs). To check this, we compared a list of 7500 genes with RPM counts of ≥10 in both replicates of Co-culture samples and having a fold-change >1 with a list of 72 genes previously shown to be overexpressed in breast cancer and 93 genes found with somatic mutations in breast cancer samples (**Axelsen et al., 2007**; **Nik-Zainal et al., 2016**). We found that mRNAs corresponding to 40 breast cancer-related overexpressed genes and 77 breast cancer-related mutated genes underwent transfer from MCF7 to MBS-MEF cells (=0.074 and p<3.776e-13, respectively, using a hypergeometric test) (**Figure 3D**; **Source data 1 - table 8**). One gene (BUB1B) was found to be both overexpressed and mutated, and to undergo transfer.

We validated the results from RNA-Seq of 10 highly transferred genes (out of the set of 283 genes mentioned above in *Figure 2D*) by qRT-PCR in two independent experiments. We also examined the transfer of four poorly transferred genes as negative controls, and β-actin as a positive control. In full agreement with the RNA-seq results (*Figure 2*), the robust transfer of RNAs shown by sequencing could also be verified using qRT-PCR (*Figure 4A*). In parallel, we validated the transfer of a set of highly transferred mRNAs using smFISH for three genes (e.g. KRT8, PSAP and CCND1) in MBS-MEF cells using human-specific FISH probes (*Figure 4B and C*). Donor MCF7cells could be readily distinguished from acceptor MEFs, due to the high levels of query mRNA expression, often presence of transcription sites, and morphological differences between the nuclei (*Figure 4—figure supplement 1*). In agreement with the RNA-seq and qRT-PCR data, we could detect robust transfer of these three genes from human MCF7 cells to mouse MBS-MEF cells (*Figure 4C*). As expected, the transfer of KRT8 mRNA was noticeably higher than that of CCND1 or PSAP, which might be attributed to the higher expression of KRT8 in MCF7 cells. Thus, we could verify intercellular RNA transfer using three different approaches.

## mRNAs encoding translation-related proteins dominate the human transferome

To check if the transferred mRNAs are associated with specific Gene Ontology terms, we analyzed the list of highly transferred mRNAs by DAVID (Database for Annotation, Visualization and Integrated Discovery) and GeneCards for the GO terms that were highly enriched (*Huang et al., 2009a*; *Huang et al., 2009b*; *Stelzer et al., 2016*). We found that most of the mRNAs were involved in translation initiation, RNA transport, ribosome biogenesis and mRNA splicing (*Figure 2—figure supplement 2—source data 1*). In terms of molecular functions, transferred mRNAs encoded for poly-A binding proteins, ribosomal proteins and translation factors and localizing to the cytosol, cell membrane or ribosome (*Figure 2—figure supplement 2—source data 1*). We also looked for pathway enrichment using KEGG (Kyoto Encyclopedia of Genes and Genomes) and Reactome databases (*Fabregat et al., 2018*; *Fabregat et al., 2017*; *Kanehisa and Goto, 2000*; *Wu and Haw, 2017*). The transferred mRNAs were enriched for pathways such as RNA transport, metabolism, translation and rRNA processing (*Figure 2—figure supplement 2—source data 1*). Since the genes encoding these cellular functions tend to be highly expressed, it might be expected that they would be the most abundantly transferred RNAs.

## Level of intercellular RNA transfer depends upon gene expression

To validate the hypothesis that gene expression is an important determinant of mRNA transfer, we employed two approaches. In the first approach, we co-cultured wild-type (WT) MEF cells with MBS-MEF donor cells expressing either high or low levels of an exogenous mRNA encoding tdTomato that was stably expressed under the control of a constitutive promoter (*Figure 5A and B*). Donor cells expressing tdTomato could be easily distinguished from acceptor cells which did not show strong red fluorescence. tdTomato-low expression cells were found to express an average 429 molecules of tdTomato mRNAs, out of which, an average of 7 molecules (~1.6%) were transferred to WT-MEF acceptor cells (*Figure 5B*). On the other hand, tdTomato-high expression cells were found to express an average of 1250 molecules of tdTomato mRNA, of which 25 molecules (2%) were able to transfer (*Figure 5B*).

In a second approach, we co-cultured WT-MEFs with donor MBS-MEFs expressing tdTomato mRNA under the control of a Doxycycline-inducible promoter in the presence of increasing concentrations of Doxycycline (*Figure 5C*). Higher concentrations of Doxycycline (4 µg/ml) induced a higher level of tdTomato mRNA expression (average = 492 copies/cell) and, consequently, transferred more mRNA to the acceptor cells (average: 1.5 copies/cell) than lower levels of Doxycycline (2 µg/ml; average expression = 165 copies per cell;<1 transferred; *Figure 5C*). Together, these results provide strong evidence of the role of gene expression in mediating mRNA transfer and in a promoter-dependent manner.

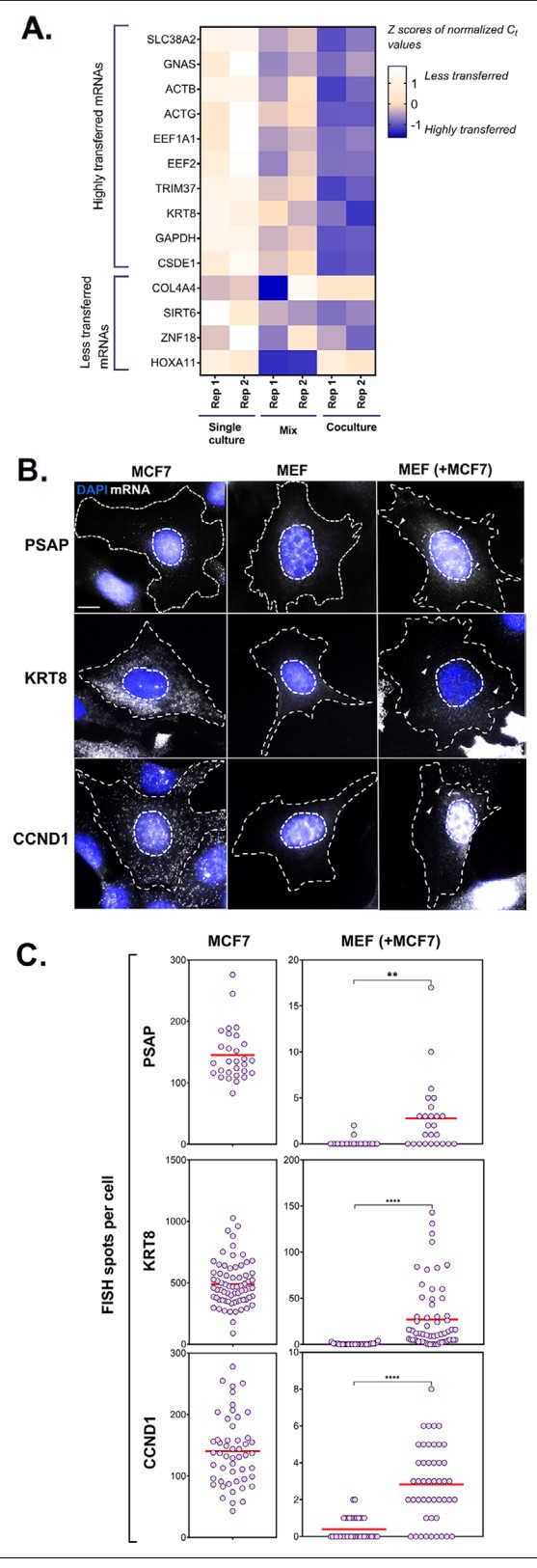

**Figure 4.** PCR and smFISH verification of transfer of selected mRNAs. (**A**) Heatmap summarizing transfer of human mRNAs in Co-culture compared to Mix. qRT-PCR was performed on RNA samples from MBS-MEF Single culture, Mix, and Co-culture samples from two independent biological replicates and the presence of transferred RNA was detected with human-specific oligos for ten transferred genes, as identified in *Figure 2D*. Four less transferred

*Figure 4 continued on next page*

*Figure 4 continued*

genes were used as negative controls. Human β-actin was used as a positive control. The color of the heatmap corresponds to the row-wise Z-scores of Ct values of the indicated genes after normalizing for an internal control (18 S rRNA). Increasing darkness corresponds to increasing gene expression. All but one gene (i.e. GNAS) showed good agreement with the RNA-seq results and have a higher fold-change as compared to the Mix sample. Each box represents an average of three independent technical replicates. (**B–C) Verification by smFISH**. Three genes (KRT8, PSAP, and CCND1) that demonstrated high level of transfer by RNA-Seq were chosen to be analyzed by smFISH. Acceptor cells (MBS-MEFs) were co-cultured with MCF7 cells together on fibronectin-coated coverslips at a ratio of 1:1 for 12 hr. Following co-culture, the cells were fixed and smFISH was performed using Quasar 570-labeled oligonucleotide probes complementary to the human-specific RNA of the indicated genes and Cy5-labeled probes specific for the MBS sequence. The transfer of mRNAs was detected by wide-field microscopy and quantified using a MATLAB program, FISH-Quant. (**B**) smFISH images. Representative smFISH images of MBS-MEF and MCF7 single cultures, and MCF7 cells in co-culture with MBS-MEFs. Labels: gray, Q570-labeled smFISH probes; blue, DAPI staining of the nucleus. Donor and acceptor cells were distinguished by the high expression of β-actin-MBS (identified by Cy5-MBS probes) in MBS-MEF cells only (not shown). Scale bar: 10 μm. (**C**) Quantification of the number of mRNAs of two independent experiments. The left panels show the number of mRNAs expressed for the indicated genes in the MCF7 cells only, while the right panel shows the number of corresponding mRNAs in MBS-MEF cells alone or in co-culture. Each dot represents the number of indicated mRNAs detected in a single cell, as measured by FISH-Quant. Horizontal red lines represent the average number of mRNAs. ** - p≤0.01; **** - p≤0.0001.

The online version of this article includes the following figure supplement(s) for figure 4:

**Figure supplement 1.** Nuclear morphology and high levels of query mRNA expression distinguish human cells from mouse cells.

## Intercellular mRNA transfer does not appear to necessitate RNA motifs

We next examined if there are any sequence motifs involved in RNA transfer. We adopted both experimental and bioinformatic approaches to examine the sequences for transfer-promoting motifs encoded either in the coding sequences or in the UTRs of RNA transferred RNAs.

To identify such an element by experimental means, we used β-actin mRNA - one of the most abundantly transferred mRNAs - as a model for RNA transfer. We reasoned that if there was a transfer-promoting element, it could enhance transfer of a reporter mRNA. We divided the coding sequence (CDS) of β-actin mRNA into three overlapping segments (CD1, CD2, and CD3) and the 3′UTR into two non-overlapping segments (3U1 and 3U2; *Figure 5—figure supplement 1A*). Each fragment was cloned downstream of the stop codon of the tdTomato reporter and the mRNA was stably expressed from a constitutive ubiquitin promoter in MBS-MEFs (*Figure 5—figure supplement 1A*). tdTomato alone (without any β-actin fragment) was used as a baseline for transfer efficiency. Transfer of the various tdTomato-β-actin constructs from MBS-MEFs to WT-MEFs in co-culture was examined by smFISH using tdTomato-specific probes (*Figure 5—figure supplement 1B*). We found that donor MBS-MEFs expressing either tdTomato alone or fused with β-actin gene fragments expressed the RNAs at similar levels (*Figure 5—figure supplement 1C*). Next, we examined RNA transfer (*Figure 5—figure supplement 1D*, *Supplementary file 1 - table 2*) and found that tdTomato mRNA alone transferred at a higher level than that of the fusion constructs, although the fusions amongst themselves showed similar levels of transfer. Thus, we could not identify a fragment of β-actin that promotes mRNA transfer, at least in the context of this assay, that is placed in the 3′UTR of tdTomato and expressed ectopically.

Next, we analyzed the 5′UTR, 3′UTR, and CDS RNA and encoded amino acid sequences of the ~280 most highly transferred mRNAs by MEME Suite (*Bailey et al., 2009*) to look for consensus motifs that might be connected to transfer (*Figure 5—figure supplement 2*). While no such motif was detected at the level of amino acid sequences or in the UTRs, three different purine-rich motifs were enriched in the coding sequences of nearly all mRNAs (*Figure 5—figure supplement 2A*). Two contain the sequence 5′–GAAGAAG-3′, which is similar to the 5′–GCAGAAG-3′ or 5′–GGAGAAG-3′ sequences present in each of the coding sequence fragments of β-actin, described above, but did not show enhanced transfer (*Figure 5—figure supplement 1D*).

To determine whether these motifs might be connected to promoting RNA transfer, we examined for the presence of motifs in genes exhibiting a low level of RNA transfer. We selected three separate sets of genes of similar number as that of the highly transferred genes (*i.e.* between 220–300 genes)

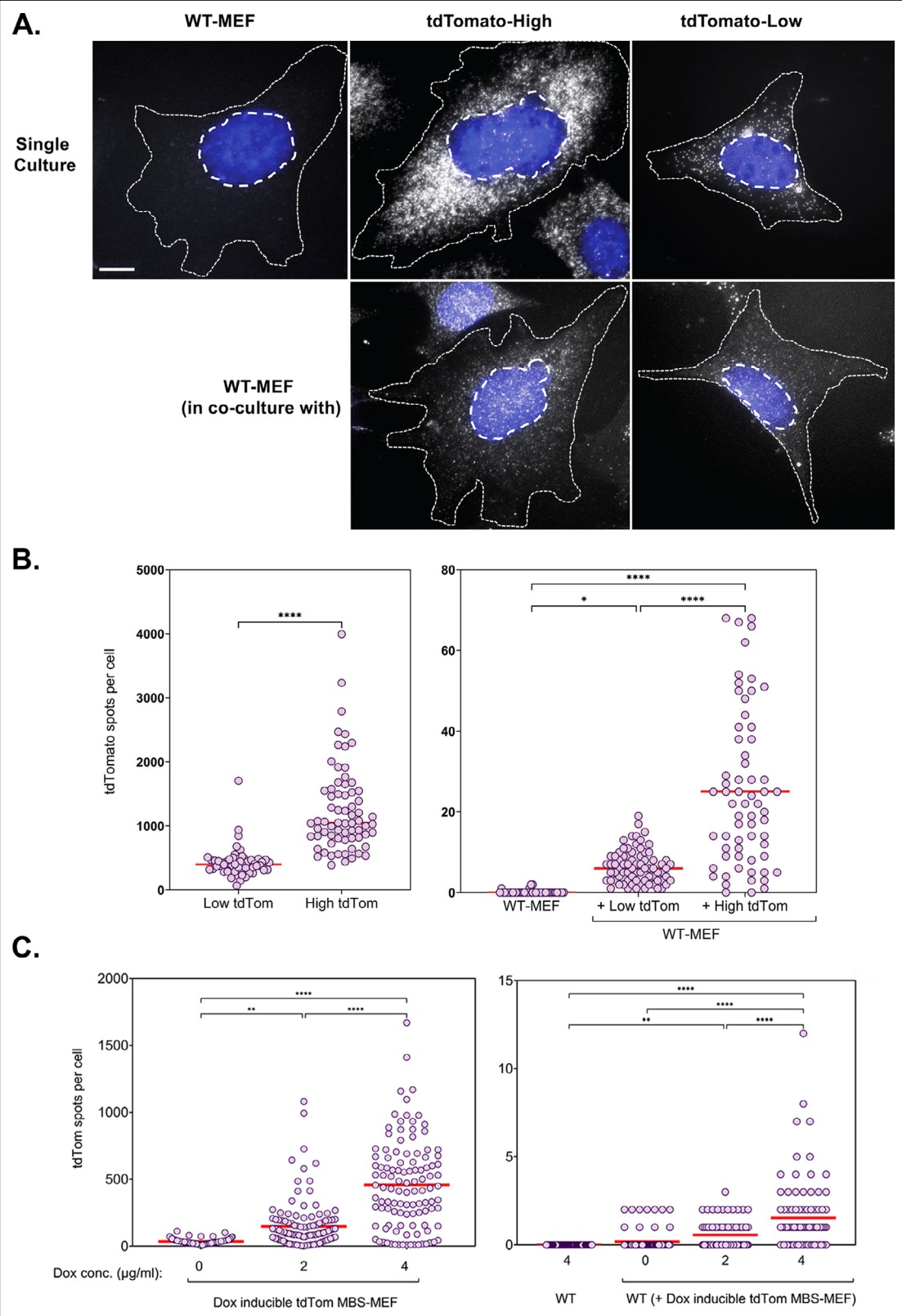

**Figure 5.** mRNA transfer is driven by gene expression. (**A**) tdTomato RNA smFISH images. Representative smFISH images of Cy5-labeled tdTomato RNA in WT MEFs and donor MBS-MEFs expressing either high or low levels of tdTomato in single cell culture (top row) and acceptor WT MEFs after co-culture (bottom row) are shown. White spots: Cy5 labeled tdTomato FISH. Blue: DAPI labeling. Dashed outlines represent the approximate cellular and nuclear boundaries. Donor and acceptor cells were differentiated by the number of spots and the presence of transcription sites, which appear only

*Figure 5 continued on next page*

*Figure 5 continued*

in donor cells. (**B**) Distribution of the tdTomato spots. FISH-Quant quantification of the number of tdTomato RNAs detected in each cell (denoted as a single dot) in donor MBS-MEFs cells (left) and WT MEF acceptor cells alone or in co-culture (right) of two independent experiments. Red horizontal line indicates the mean of the distribution. The samples in WT-MEF and in co-culture were compared by a one-way ANOVA, followed by indicated post-hoc pairwise comparisons. Expression in the two donor cell populations were compared by unpaired t-test. * - p≤0.05; **** - p≤0.0001. (**C**) Use of a doxycycline-inducible tdTomato. MBS-MEFs stably expressing tdTomato under the control of a doxycycline-inducible promoter were incubated with increasing concentrations of doxycycline (0–4 mg/ml) for 24 hr prior to co-culture with WT MEFs in medium containing the same concentration of doxycycline. Left panel: Score of smFISH labeling of donor tdTomato MBS-MEFs using probes against tdTomato RNA. Right panel: Score of smFISH labeling of tdTomato RNA in acceptor WT MEFs after co-culture. Summary of two independent experiments. Expression (left panel) and RNA transfer (right panel) between untreated and doxycycline-treated cells, respectively, was compared using an unpaired t-test. Red horizontal line indicates the mean of the distribution. * - p≤0.05; ** - p≤0.01; *** - p≤0.001; **** - p≤0.0001.

The online version of this article includes the following source data and figure supplement(s) for figure 5:

**Figure supplement 1.** No *cis* element is involved in β-actin mRNA transfer.

**Figure supplement 2.** MEME analysis of transferred genes.

**Figure supplement 2—source data 1.** This file contains three lists of low-transferred mRNAs and their transfer-related data used for MEME analysis in *Figure 5—figure supplement 2*.

(*Figure 5—figure supplement 2—source data 1*) and looked for consensus sequences in the coding regions (*Figure 5—figure supplement 2B*). Interestingly, all three sets contained similar purine-rich sequences as found in the highly transferred genes with the 5′–GAAGAAG-3′ motif identifiable in Sets 2 and 3 (*Figure 5—figure supplement 2B*). As this sequence is not specific to highly transferred genes it is unlikely to be a transfer-promoting feature.

Short motifs have been identified in EV-sorted miRNAs (*Garcia-Martin et al., 2022*; *Santangelo et al., 2016*; *Temoche-Diaz et al., 2019*) and mRNAs (*Batagov and Kurochkin, 2013*; *Bolukbasi et al., 2012*; *Szostak et al., 2014*). We therefore tested whether motifs found in EV-sorted mRNAs could be found in our robustly transferred mRNA cohort. However, we found that one motif is completely absent from this cohort, as well as from our three control cohorts described above. The other four motifs were found in only 10–19% of the robustly transferred mRNAs, but were present at a similar level (10–25%) in our control sets (*Supplementary file 1 - table 3*). Thus, none of the three approaches could conclusively identify *cis* elements that might promote RNA selection for transfer.

## Intercellular mRNA transfer depends on direct cell-to-cell contact

Multiple studies have reported the presence of RNAs in extracellular vesicles, such as exosomes and exosome-like vesicles (*Ekström et al., 2012*; *Matsuno et al., 2019*; *Valadi et al., 2007*). More importantly, breast carcinoma cell lines such as the MCF7 cells used in this study have been shown to release exosomes and other EVs that can package RNAs and influence the physiology of acceptor cells (*Jafari et al., 2020*; *Lau and Wong, 2012*). In contrast to EV-mediated transfer, however, we demonstrated that mRNAs transfer via TNTs that confer direct cell-cell contact, and not by diffusion through the media (*Haimovich et al., 2017*; *Haimovich and Gerst, 2019*). The transfer of both β-actin and GFP mRNAs was completely abrogated when donor and acceptor cells were separated in space (e.g. using a tripod/transwell setup) or when the acceptor cells were treated with donor cell-derived conditioned medium (*Haimovich et al., 2017*). Thus, we asked whether the human mRNAs detected in mouse acceptor cells were contributed by a contact-dependent mechanism (e.g. TNTs) or via contact-independent pathways (e.g. EVs). To test this, we co-cultured MCF7 donor cells and MBS-MEF acceptor cells on the same plate or physically separated several millimeters apart using a quadrapod plate setup (see *Materials nd methods*). Alternatively, we treated the MBS-MEF cells with conditioned medium derived from a MCF7 single culture (*Figure 6—figure supplement 1A*). We then separated the cells and extracted RNA as described above and followed the transfer of select mRNAs by qRT-PCR. We chose four genes from the list of highly transferred genes identified by RNA-sequencing (e.g. ACTG, GAPDH, TRIM37, and KRT8), while using ACTB as a positive (TNT-dependent) control for the analysis. The transfer of mRNAs was found to be highest in the Co-culture samples as compared to the quadrapod or conditioned media-treated samples across two independent replicates (*Figure 6—figure supplement 1B*). This indicates that the mRNAs did not transfer by diffusion (e.g. either by secreted vesicles or ribonucleoprotein complexes), but requires cell-cell contact. As the genes selected for this experiment encode diverse biological functions and do not belong to

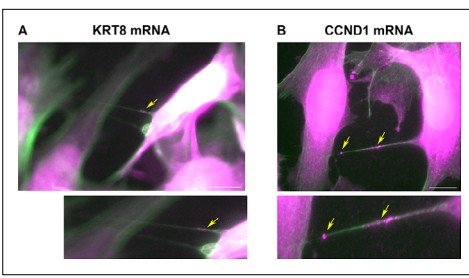

**Figure 6.** Cells are connected by mRNA-containing TNTs. (**A**) KRT8 FISH image of MCF7-MEF co-cultures show actin-based TNTs (green) that contain KRT8 FISH spots (magenta). Shown is a single z slice of the FISH image, and an enlarged image of the TNTs. Brightness was amplified to allow easy visualization of the FISH spots (indicated by yellow arrows) in TNTs. Scale bar: 10 μm. See *Figure 6—video 1* for the whole z stack. (**B**) CCND1 FISH image of MCF7-MEF co-cultures show actin-based TNT (green) that contain CCND1 FISH spots (magenta). Brightness was amplified to allow easy visualization of the FISH spots (indicated by yellow arrows) in TNTs. Shown is a max-projection of three middle z slices from the FISH image, and an enlarged image of the TNT. Scale bar: 10 μm. See *Figure 6—video 2* for the whole z stack.

The online version of this article includes the following video and figure supplement(s) for figure 6:

**Figure supplement 1.** Role of cell-cell contact in intercellular mRNA transfer.

**Figure supplement 2.** MEFs and MCF7 cells are connected by TNTs.

**Figure supplement 3.** Cells are connected by mRNA-containing TNTs.

**Figure 6—video 1.** Full z stack of *Figure 6A*.
https://elifesciences.org/articles/83584/figures#fig6video1

**Figure 6—video 2.** Full z stack of *Figure 6B*.
https://elifesciences.org/articles/83584/figures#fig6video2

any common pathway, it suggests that contact-dependent transfer is likely to be the predominant mechanism for transcriptome-wide mRNA transfer.

## Intercellular mRNA transfer occurs via TNTs

We previously showed that mRNA transfer is mediated by TNTs (*Haimovich et al., 2017*; *Haimovich and Gerst, 2019*). In order to verify that TNTs are formed by the two cell types, MEFs and MCF7, we created stable cell-lines of MEFs and MCF7 expressing either palmytoilated GFP or TagRFP-T (GFP-ps or TagRFP-T-ps), which locate to cell membranes, and looked for TNTs formation by live imaging. We found that both cell types produce TNTs which connect between MEFs-MEFs, MCF7-MCF7 and MEFs-MCF7 cells (*Figure 6—figure supplement 2*). Next, we searched through our FISH images for TNTs. TNTs are fragile and are destroyed easily during the FISH protocol (*Haimovich and Gerst, 2019*) and are therefore difficult to detect in FISH images. Nevertheless, we found a few examples of MCF7 cells connected by TNTs to other MCF7 cells (*Figure 6—figure supplement 3A, B*), or to MEFs (*Figure 6—figure supplement 3C–E*). Most of these TNTs contained FISH spots that represent KRT8 mRNA (*Figure 6—figure supplement 3A–C*) or CCND1 mRNA (*Figure 6—figure supplement 3E*). To further characterize these structures, we repeated the smFISH experiments for KRT8 and CCND1, with the addition of phalloidin-FITC, which stains actin filaments in green. *Figure 6* and *Figure 6—video 1* and *Figure 6—video 2* show two examples of KRT8 or CCND1 mRNAs in actin-containing long protrusions that run above the substratum and connect two cells – key features of TNTs (*Ljubojevic et al., 2021*). Together with the results presented in *Figure 6—figure supplement 1*, we conclude that mRNA transfer in our co-culture system likely occurs via TNTs.

## Co-culture with human cells leads to differential changes in the mouse transcriptome

Next, we were curious to check the impact of co-culture conditions on acceptor cells. RNA-seq analysis revealed that the mouse transcriptome in Co-culture samples was significantly different from that of the Mix and Single cultures, which cluster together and have very similar transcriptomic profiles (*Figure 7A and B*). This indicates that MBS-MEFs undergo stark changes in gene expression upon co-culture with MCF7 cells. We detected more than 4000 genes that are ≥2 fold differentially expressed, including ~1000 upregulated genes and ~3000 downregulated genes (*Figure 7C*) (*Source data 1 - table 9*). Interestingly, we observed the ~3.5-fold upregulation of a cancer-associated fibroblast (CAF) marker, Tenascin-C (TNC) (*Ni et al., 2017*), in the MEF-enriched fraction of the Co-culture sample (*Source data 1 - table 9*). This raises the speculation that MEFs might become CAF-like when co-cultured with a cancer cell line (e.g. MCF7 cells). To explore this further, we compared the

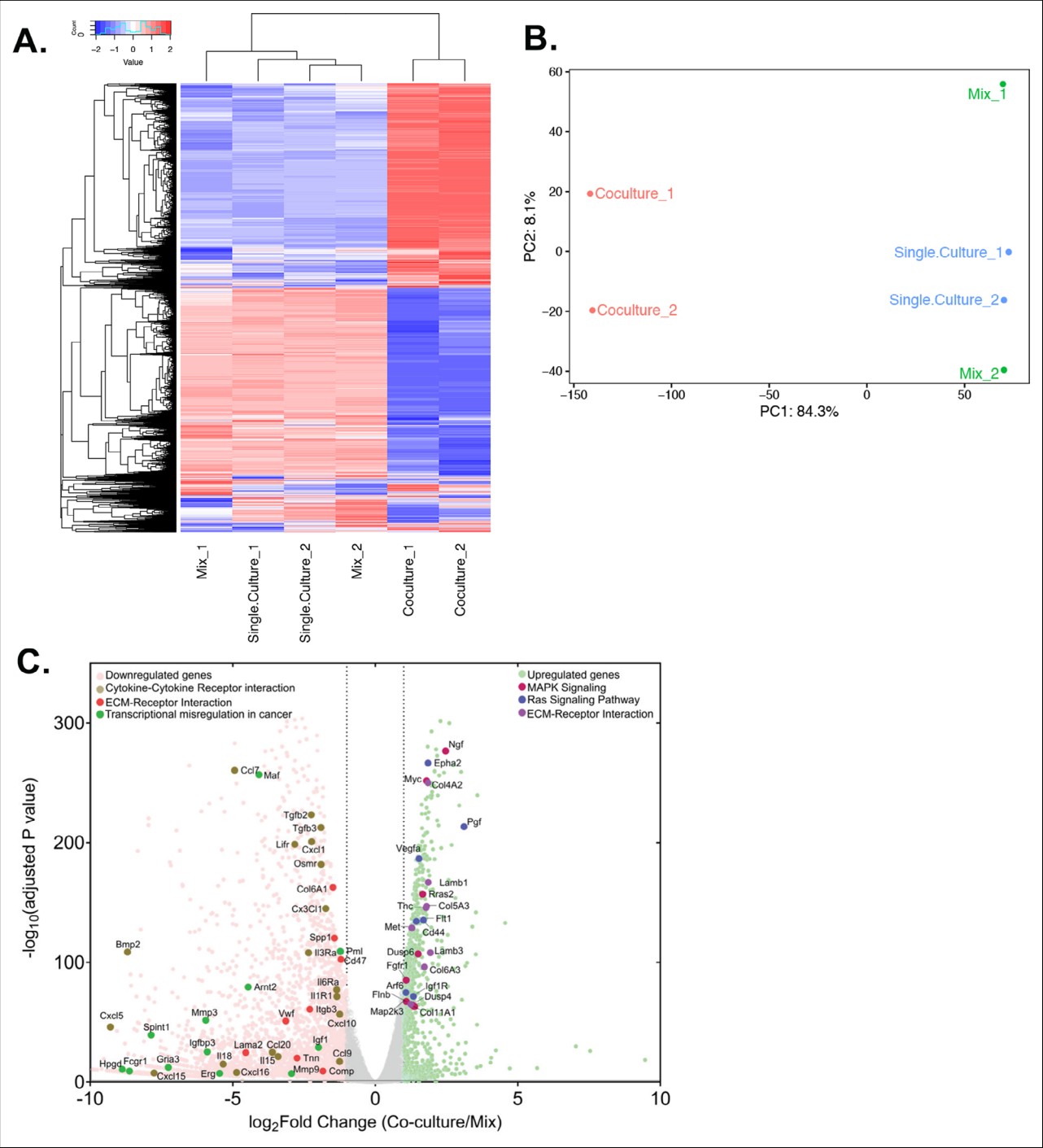

**Figure 7.** Differential gene expression in acceptor cells in response to co-culture. RPM counts from the mouse fraction of MBS-MEF Single culture, Mix, and Co-culture samples were aligned with the mouse reference genome (mm10) and analysed further as follows. (**A**) Heatmap analysis indicating up-and down-regulated clusters of genes. Normalized read counts (RPM) of mouse-specific genes across Single culture, Mix, and Co-culture mouse samples were Z-normalized across rows and plotted using G-Plot package of R. Each row indicates a single gene and each column represents a sample. Cluster analysis of the samples was done by 'maximum' distance function and the top dendrogram shows the results. (**B**) PCA analysis of the samples: Mouse-specific reads from mouse samples were analyzed and plotted, as a Principal Component Analysis graph by using the base function of R. Red icon: Co-culture, Blue icon: Single Culture and Green icon: Mix. (**C**) Volcano plot depicting differentially regulated genes and representative pathways: Mouse-aligned genes in two replicates of 'Mix' and 'Co-culture' were analyzed for differential gene regulation by Deseq2 package in R with default parameters. Results are shown as a volcano plot with the fold change in co-culture over mix and the corresponding adjusted p-values. Threshold limit for significance was set at $P$=0.05. Upregulated genes are shown in light green, while the downregulated genes are shown in pink. Gray dots represent

*Figure 7 continued on next page*

*Figure 7 continued*

statistically insignificant genes. Black dashed lines show the fold change threshold value of ±2. Selected top upregulated and downregulated genes grouped by their cancer-related KEGG pathways are indicated.

The online version of this article includes the following source data and figure supplement(s) for figure 7:

**Figure supplement 1.** Analysis of differentially expressed genes in acceptor MEFs after co-culture.

**Figure supplement 1—source data 1.** This file contains GO and KEGG terms analysis of MEF endogenous mRNAs that were up or down regulated in co-culture.

---

fold-change of differentially regulated genes from MEFs grown in Co-culture (versus those from the Mix) with the corresponding genes in CAFs (versus their precursor mesenchymal stem cells - MSCs), as reported in a recent study (*Yu et al., 2020*). Interestingly, we found 30 upregulated and 17 down-regulated genes conserved between the two datasets (*Figure 8A*, *Source data 1 - table 10*). Among upregulated genes in the MEFs, multiple extracellular-matrix associated genes (e.g. ACTA1, ACTA2, COL6A3, ADAM12, ADAM19) were also upregulated in CAFs associated with metastatic lung cancer (*Figure 8B*). Furthermore, mouse KRT8, which is frequently associated with apical CAFs (apCAFs) in the breast and pancreatic cancer microenvironment (*Elyada et al., 2019*; *Sebastian et al., 2020*) was upregulated in MEFs in response to co-culture with MCF7 cells (*Source data 1 - table 9*). This indicates that co-culture with human cancer cells might lead to CAF-like phenotypes in cultured fibroblasts.

We analyzed the differentially expressed mouse genes for upregulated and downregulated pathways using KEGG. Top upregulated pathways included 'Ribosome biogenesis in eukaryotes', 'RNA transport' and 'Focal adhesion', while top downregulated pathways included signaling pathways, such as 'Calcium Signaling pathway' and 'Rap1 signaling pathway' (*Figure 7—figure supplement 1*, *Figure 7—figure supplement 1—source data 1*). Upon close observation, many cancer-related pathways (e.g. MAPK signaling, Ras signaling) were found to be upregulated, while pathways such as cytokine-cytokine interactions and transcriptional misregulation in cancer were downregulated (*Figure 7C*).

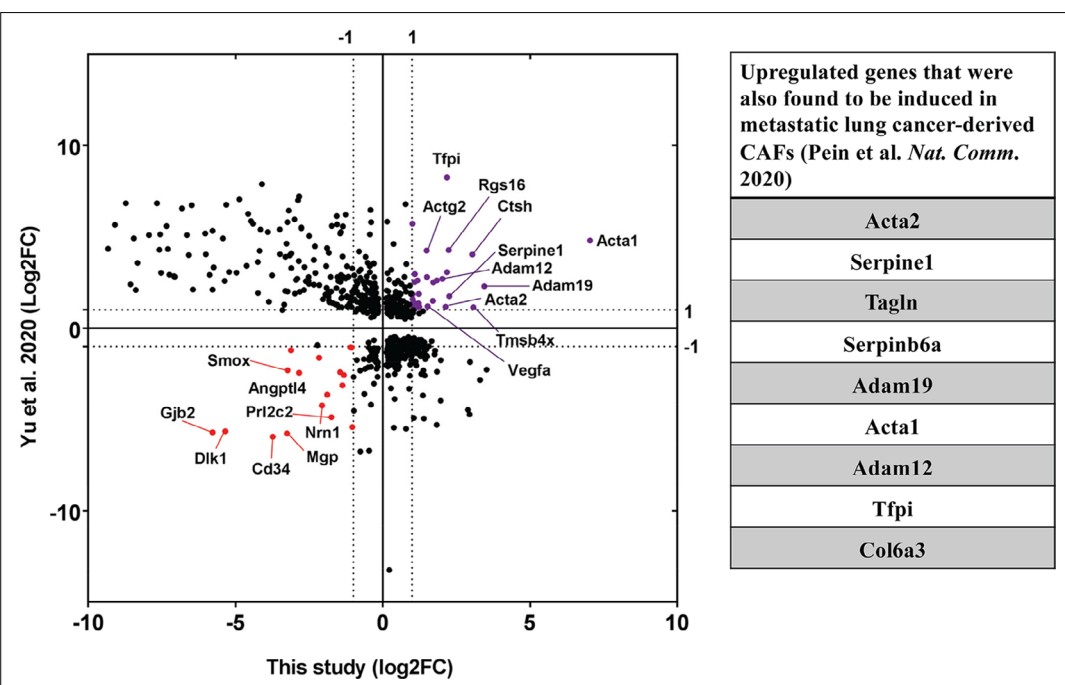

**Figure 8.** Upregulation of CAF-associated genes in MEF acceptor cells in co-culture. Fold changes of differentially regulated genes ($p_{adj}$ <0.05) in MEFs in co-culture compared to MEF in Mix (*Figure 7*) were compared to fold changes of corresponding genes in EG7-tumour derived CAFs over their precursor Mesenchymal Stem Cells (MSC) (*Yu et al., 2020*). Each point represents a gene. Thirty genes (marked in purple) were found to be significantly upregulated in both studies while seventeen genes (marked in red) were found to be downregulated. Among the upregulated ones, several genes (listed in the table on the right) were found to be upregulated in metastatic lung cancer-derived CAFs (*Pein et al., 2020*).

---

Another major cancer-related pathway, 'ECM-Receptor interaction' was found to be both up- and downregulated in MBS-MEFs in response to co-culture with MCF7 cells (*Figure 7C*), with most of the upregulated genes associated with poly-A RNA binding, translation initiation, ribosome biogenesis and ribosome assembly-related GO terms (*Figure 7—figure supplement 1*). Interestingly, cell immunity-related genes were significantly downregulated, leading to speculation that RNA transfer and/or co-culture modulates the immune response of acceptor cells (*Figure 7—figure supplement 1D*).

## Discussion

In this study, we employed a genome-wide transcriptomic-based approach to understand the global prevalence of intercellular mRNA transfer. Using a human-murine co-culture system, we developed a simple, comprehensive, and quantitative method to study RNA transfer that involves co-culturing heterologous cell populations, sorting them by antigen-based affinity purification, and performing deep sequencing (*Figure 1A*). We found that numerous RNAs undergo transfer (*Figure 2A* and *Source data 1 - tables 1 and 2*) and that the number of transferred mRNAs strongly correlates linearly with the endogenous expression level in donor cells (*Figure 2F–G*). This, in turn, led to the discovery that most abundantly transferred RNAs are highly expressed genes such as β- and γ-actin, cytoskeletal components, translation factors, ribosomal subunits and other genes (*Figure 2D*). We found that the level of gene expression alone explains more than 90% of RNA transfer (*Figure 2E–G* and *Figure 5*). This is in agreement with our previous finding that transfer of human Cyclin D1 mRNA and mouse β-actin mRNA correlates with the expression level in donor cells (*Haimovich et al., 2017*). Another study, that reported transfer of keratinocyte-associated mRNAs to Langerhans cells, mostly through a contact-dependent mechanism, also found that the amount of transferred RNAs (as quantified by PCR) positively correlated with gene expression (*Su and Igyártó, 2019*).

Similar results have also been reported in plants, where the long-distance transport of mRNAs between non-contiguous cells is achieved via plasmodesmata and sieve elements that play a major role in development. An analysis of mRNAs exhibiting long-distance transport in *Arabidopsis* showed that was largely non-selective and dependent on gene expression (*Calderwood et al., 2016*; *Hofmann, 2016*). In case of RNA transfer between species, such as host-parasitic plants, a large proportion of host and parasite mRNAs were found in each other's tissues in an expression-dependent manner (*Notaguchi et al., 2015*). In an instance of cross-species grafting, no specific *cis* motif was found among *Arabidopsis* mRNAs that were found in *Nicotiana* tissues (*Notaguchi et al., 2015*). However, a recent report found that RNA modification [5-methyl cytosine ($m^5C$)] mainly in the proximal end of coding regions was responsible for transport of certain mRNAs from shoot to root of *Arabidopsis* (*Yang et al., 2019*). In addition, two works describe a tRNA-like structure or actual bi-cistronic mRNA-tRNA transcript that acts as a *cis*-acting element in plant mRNA transfer (*Huang and Yu, 2009*; *Zhang et al., 2016*).

We did not detect any gene with a disproportionally high level of transfer with respect to endogenous expression (*Figure 2D–G*), indicating that the presence of other transfer promoting factors (e.g. RNA motifs; *Figure 5—figure supplements 1 and 2*, and *Supplementary file 1 - table 3*) beyond gene expression is unlikely. Short motifs have been identified in EV-sorted miRNAs (*Garcia-Martin et al., 2022*; *Santangelo et al., 2016*; *Temoche-Diaz et al., 2019*) and mRNAs (*Batagov and Kurochkin, 2013*; *Bolukbasi et al., 2012*; *Szostak et al., 2014*). Insertion of one such motif into a reporter RNA even influenced its recruitment into EVs (*Szostak et al., 2014*). At this point, the issue of full-length mRNAs in EVs remains somewhat controversial and therein represent only a small fraction of the native transcriptome and are present at levels that do not correspond necessarily to their level of transcription (reviewed in *Prieto-Vila et al., 2021*). While such motifs may be functional, we could not identify any associated specifically with mRNA/lncRNA transfer using different approaches (*Figure 5—figure supplements 1 and 2*, and *Supplementary file 1 - table 3*). On the other hand, we showed previously that the level of contact-dependent mRNA transfer can depend on cell type, as HER2 mRNA transfer from two human cell lines (e.g. NCI-N87 and SKBR3 cells) to MEFs was similar, although the expression level in SKBR3 cells was 2-fold higher (*Haimovich et al., 2017*). Thus, RNA transfer might correlate with endogenous expression levels within the same cell type, but not necessarily across different cell types. The existence of transfer-inhibiting or -promoting elements recognized by cell-type specific factors could provide a possible explanation why some cell types transfer

certain mRNAs or lncRNAs better or worse than others, even when taking gene expression levels and growth conditions (e.g. stress conditions or presence of extracellular matrix proteins, like fibronectin) into consideration. These hypotheses should be further explored, since cell type-derived differences in RNA transfer efficiency could have different physiological outcomes.

Interestingly, a report characterizing the transcriptome of the EV populations derived from hepatic cancer cell lines identified 238 protein-coding mRNAs and 35 lncRNAs, in addition to numerous miRNAs and snoRNAs (*Berardocco et al., 2017*). Many of these mRNAs were found to be highly transferred here and included genes encoding ribosomal proteins, ACTG, ACTB, and EEF2, *etc*. Along with an earlier work (*He et al., 2015*), it implies that mRNAs or mRNA fragments might be transferred by hepatic EVs and that the loading of RNAs into such vesicles may be expression-dependent. However, in this study and in our previous work (*Haimovich et al., 2017*) we could not detect diffusion-based mRNA transfer. In contrast, we now provide strong evidence that transfer of a few example mRNAs is contact dependent and likely occurs via TNTs (*Figure 6* and *Figure 6—figure supplements 1–3*).

Besides expression dependence, another hallmark of intercellular RNA transfer is the relatively low level of transfer (e.g. average of <1% of endogenous gene expression; *Figure 3A*). This makes transfer difficult to detect by bulk RNA sequencing and even more so by single-cell RNA sequencing (scRNA-seq). scRNA-Seq platforms, such as DropSeq, have a low level of accuracy and the sensitivity drops further when analyzing low expressed genes (e.g. <50 copies of mRNA per cell) (*Torre et al., 2018*). In addition, data from scRNA-seq using low amounts of input RNA is noisy and, thus, is less reliable (*Brennecke et al., 2013*). In an attempt to detect transferred RNAs from MCF7 to MBS-MEFs using a newly described scRNA-Seq process, BAG-Seq (*Li et al., 2020*), only ~2000 genes were identified, yet none of which (not even β-actin) could be detected as having undergone transfer (data not shown). Further improvements in scRNA-seq technology, in terms of sensitivity, might go far towards allowing exploration of this phenomenon especially in tissues. Alternatively, multiplex FISH approaches (e.g. MERFISH or seqFISH+ *Eng et al., 2019*; *Xia et al., 2019*) might prove superior to scRNA-seq due to higher sensitivity. In fact, our smFISH results (*Figure 4B and C*) suggest that RNA-seq undercounts RNA transfer. For example, the average percentage of transfer of ACTB, KRT8, PSAP and CCND1 mRNA as assayed using FISH is 5–10 fold higher than the estimate from RNA-seq (i.e. compare data in *Figure 4C* to that in *Source data 1 - table 4*). This difference could be either methodological or technical in origin, as smFISH measures the mRNA in unperturbed adherent cells, whereas for RNA-seq the cells are first detached from the surface by trypsin at 37 °C and then sorted (at 4 °C) prior to RNA extraction. We have observed that the trypsinization and re-plating of acceptor MEFs leads to a massive loss in transferred ACTB-MBS mRNA, possibly through degradation (data not shown). On the other hand, RNA-seq library preparation may introduce bias in the detection of high abundance transcripts versus lower abundance ones at the reverse transcription and/or amplification levels. Thus, there is an experimental bias towards the detection of endogenous mRNAs, as compared to transferred mRNAs, and multiplex FISH methods may eventually prove more accurate. We note that the duration of our co-culture experiments was 12 hr and it is possible that longer durations might reveal a higher percentage of transfer. Although the transfer of β-actin-MBS mRNA reached a maximum within 2.5 hr, and was maintained at the same level for at least 24 hr when either immortalized or primary MBS-MEFs were co-cultured with immortalized or primary wild-type MEFs (*Haimovich et al., 2017*), the transfer of β-actin-MBS mRNA continued to increase at least for up to 12 hr when MBS-MEFs were co-cultured with MCF7 cells (*Dasgupta and Gerst, 2020*). Thus, greater levels of accumulation of transferred RNAs might occur depending upon cell type and growth conditions, as well as the RNA transfer rate and stability in the acceptor cells.

Our results imply that nearly all human mRNA species, as well as some lncRNAs, can undergo transfer, yet what parameters limit the level of transfer, aside from relative gene expression, is unknown. As mRNAs and lncRNAs are localized to specific regions/organelles in cells (*Buxbaum et al., 2015*; *Cabili et al., 2015*), it may depend upon the ability of an RNA to localize, diffuse, or be targeted along with organelles to TNT entrance sites. Although mitochondria were shown to transfer through TNTs (*Rustom et al., 2004*; *Sartori-Rupp et al., 2019*; *Wang et al., 2011*; *Zou et al., 2020*), our RNA-seq results showed poor transfer of mitochondria-encoded RNAs relative to their expression (*Figure 2—figure supplement 1* and *Source data 1 - table 4*). It may be that TNT-mediated mitochondrial transfer is limited using our experimental conditions or that it may not be an efficient vehicle for transfer. On the other hand, limited live imaging data suggests that transfer is motorized

(*Haimovich et al., 2017*) and, thus, the RNA might require specialized packaging or modification for transfer and recognition by TNT-specific motor proteins. This packaging or modification may distinguish RNAs targeted for transfer from non-transferred RNAs (*Haimovich and Gerst, 2019*). Thus, it is tempting to speculate that RNA molecules may be co-transcriptionally 'marked' for transfer, especially since mRNA fate may be determined during transcription (*Dahan and Choder, 2013*; *Haimovich et al., 2013*; *Nair et al., 2021*; *Slobodin and Dikstein, 2020*; *Trcek and Singer, 2010*). Alternatively, transfer could be isoform-specific. However, the low read counts of transferred RNAs in our RNA-seq experiment may have prevented us from determining if they are enriched for specific transcription start sites, alternative poly-adenylation sites, or alternatively spliced isoforms. Understanding the mechanism of mRNA transfer will be key to further explore its physiological importance.

We validated several examples of transferred mRNAs that transfer via a contact-dependent mechanism, likely TNTs (*Figure 6* and *Figure 6—figure supplements 1 and 3*), and extrapolate from them to the entire transcriptome. Although it is possible that some or many mRNAs transfer by means other than TNTs, we think it unlikely, since the results on TNT-mediated cell-to-cell transfer in both this and our previous publication (*Haimovich, 2017*), as well as by others (*Ortin-Martinez et al., 2021*; *Su and Igyártó, 2019*), tested a variety of mRNAs from different families and which localize to various subcellular localizations. This indicates that the pathway we have uncovered is more general than the few examples presented here. Of note, our key findings were recently recapitulated in a different human-mouse co-culture system. In this *bioRxiv* pre-print (*Yingying et al., 2023*), the authors co-cultured human pluripotent stem cells with mouse epiblast stem cells. The cells were then FACS-sorted to human and mouse populations and either bulk RNA-seq or scRNA-seq was performed. In both cases, the authors found a contact-dependent human-to-mouse mRNA transfer, at levels ranging from 0.33% to 1.3% for bulk RNA-seq and 1.2% for scRNA-seq, similar to our findings presented here. Likewise, mRNA transfer was found to be non-specific and the rate of transfer correlated to the level of gene expression in the human donor cells. Although these experiments did not include a 'Mix' control, which is essential to remove background, nor validated by FISH, these independent results, if true, strengthen our findings.

The findings of this study will enable us to answer if TNT-mediated RNA transfer plays a role in key biological processes, such as tumor growth, tissue differentiation and development as investigated for EVs (*Liu et al., 2018*; *Prieto-Vila et al., 2021*; *Raulf et al., 2018*). Our experimental system can be considered to be a simplistic recapitulation of a two dimensional human xenograft cancer model, where the human tumor cells (MCF7s) are in close proximity with mouse fibroblasts (MEFs). Based on our results, we speculate that fibroblasts in vivo could potentially acquire CAF-like phenotypes upon co-culture, as indicated by the transfer of prooncogenic RNAs to the MEFs (*Figure 3D*), as well as dramatic changes to the MEF transcriptome incurred upon co-culture (*Figures 7 and 8*, and *Figure 7—figure supplement 1*). While at the moment we cannot exclude the possibility that the latter changes result from additional signaling pathways (e.g. paracrine, adhesion-dependent signaling, *etc.*), the idea that TNT-mediated RNA transfer could play a significant role should be considered. Although performed in vitro, our findings with immortalized cells may translate to in vivo systems, whereby tumorigenic or other signal-inducing RNAs transfer from cancer cells to surrounding stromal cells, CAFs, or immune cells. TNTs have been detected in multiple cancer types, such as bladder carcinoma, urothelial carcinoma, breast cancer, cervical carcinoma, colon cancer, glioblastoma, etc. (reviewed in *Roehlecke and Schmidt, 2020*). TNTs can then transport cargo to other cancer cells, which can increase intra-tumor heterogeneity or connect with cells in the tumor microenvironment to induce tumorigenic pathways. It is well known that RNAs derived from acute myelogenous leukemia (AML) cells make the bone marrow niche more permissive to tumor growth and evade chemotherapy (*Hornick et al., 2016*; *Huan et al., 2015*; *Kumar et al., 2018*). Furthermore, TNTs were shown to transport mutant forms of KRAS to induce ERK signaling in colorectal cancer cells and multidrug resistance P-glycoprotein in multiple cancer types (*Ambudkar et al., 2005*; *Desir et al., 2019*).

Aside from the tumor microenvironment, TNT-mediated RNA transfer probably has physiological functions under other conditions. We and others previously found that cellular stress conditions modulate TNT formation (*Ariazi et al., 2017*) and mRNA transfer (*Haimovich et al., 2017*). Thus, TNTmediated RNA transfer may have a role in maintaining tissue homeostasis or signaling under stress conditions. A particular example is the role of TNT in ocular homeostasis and pathology (*Chinnery and Keller, 2020*), and the recent discovery of TNT-mediated mRNA transfer in the retina

(*Ortin-Martinez et al., 2021*). However, more work is needed to establish the role of TNT-mediated RNA transfer in vivo. Key to understanding the significance of this process is to determine the fate of transferred mRNAs, in terms of their translation, stability, and impact upon cell physiology.

Finally, it has not escaped our imagination that contact-dependent RNA transfer mechanisms could eventually be used as a novel strategy to deliver mRNA-based drugs in vivo (*Kowalski et al., 2019*; *Sahin et al., 2014*; *Tang et al., 2019*; *Van Hoecke and Roose, 2019*; *Weng et al., 2020*; *Zhang et al., 2019a*). Current strategies to administer in vitro transcribed mRNAs inside the body using transfection or encapsulation inside lipid nanoparticles are either ineffective or elicit potent immune responses and, thus, it may be feasible to employ exogenous cells as vehicles for gene therapy (*Sahin et al., 2014*).

## Materials and methods

### Plasmids, cells and cell line generation

tdTomato was subcloned from a previously described plasmid (Addgene plasmid # 85453; gift of Aviv Regev, Broad Institute, MA) (*Singer et al., 2016*) into a pHAGE-UBC-GFP lentiviral vector behind a constitutive expression Ubiquitin C (UBC) promoter (obtained from R.H.S), and replacing the GFP, leading to the creation of the pUBC-tdTomato plasmid (Addgene plasmid # 183710). Fragments of β-actin were RT-PCR amplified (see *Supplementary file 2* for primers) from cDNA derived from WT-MEFs and cloned after the stop codon of the tdTomato ORF of pUBC-tdTomato plasmid by restriction-free (RF) cloning (*van den Ent and Löwe, 2006*) (Addgene plasmids # 183714, 183715, 183716, 183717). TagRFP-T-ps (palmitoylated TagRFP-T) was previously described *Haimovich et al., 2017*; Addgene plasmid #178656. We used RF cloning to create GFP-ps (Addgene plasmid # 183712) by subcloning the palmitoylation signal from TagRFP-T-ps to the N-terminus of GFP in pUBC-GFP.

To create the inducible tdTomato plasmid (Addgene plasmid # 183718), a lentiviral plasmid encoding the KRAB-dCas9-EGFP (pLV hU6-sgRNA hUbC-dCas9-KRAB-T2a-GFP) (Addgene plasmid # 71237; gift from Charles Gersbach, Duke University, NC) (*Thakore et al., 2015*) was used as backbone to generate a Dox-inducible tdTomato system. Briefly, the UBC promoter was replaced by a TRE3GV promoter (taken from Addgene plasmid # 85556; gift from Eric Lander, Broad Institute, MA; *Fulco et al., 2016*), the KRAB-dCas9-EGFP was replaced by tdTomato, and the U6-sgRNA cassette was replaced by a rTet cassette under an UBC promoter (taken from Addgene plasmid # 50917; gift from Rene Maehr and Scot Wolfe, University of Massachusetts Medical School, MA; *Kearns et al., 2014*). All modifications were done by RF cloning.

Lentivirus particles were produced by transiently transfecting the expression plasmid with packaging plasmids VSVG, RRE and Rev (Addgene plasmids # 12259, 12251, and 12253, respectively; gift from Didier Trono; EPFL, Switzerland) (*Dull et al., 1998*) into HEK293T cells using TransIT-Lenti transfection reagent (Mirus Bio) and allowed to grow for 72 hr. The virus-containing media were harvested and concentrated with Lenti-X concentrator (Clontech) per the manufacturer's instructions. Virus particles were resuspended in complete DMEM, aliquoted, and stored in –80 °C.

MCF7 cells were obtained from ATCC. Immortalized MEFs and MBS-MEFs cell lines were derived from mice as described in *Lionnet et al., 2011*. All cell lines tested Mycoplasma negative by PCR (*Dasgupta and Gerst, 2019*).

For all stable cell line generations (e.g. MBS-MEFs expressing either tdTomato-β-actin fragments or tdTomato expressed under the doxycycline-inducible promoter, MEFs and MCF7 expressing either TagRFP-T-ps or GFP-ps), 50,000 MBS-MEFs or MCF7 cells were seeded in ix-well plates and exposed to the virus particles mentioned above in serum-free media supplemented with 6 μg/ml polybrene (Sigma). Cells with high or low expression of tdTomato were selected by FACS (BD Biosciences Aria III).

### Cell sorting using magnetic microbeads

We used an antigen-based cell sorting protocol slightly modified from the one we described earlier (*Dasgupta and Gerst, 2020*). Culture plates (15 cm) were coated with 10 μg/mL of Fibronectin (FN) (Sigma) in PBS for 20 min before plating the cells. For the Mix and Single culture samples, MBS-MEF and MCF7 cells were cultured for about 15–18 hr on FN-coated plates before harvesting. For the Co-culture samples, MCF7 cells were plated first. Between 8 and 10 hours later, the MBS-MEF

cells were plated, and co-culturing was maintained for additional 12 hr. At the time of harvesting, cells were trypsinized using 3 ml of 0.25% Trypsin (Sigma) and resuspended in 500 µl of cold DMEM (supplemented with FBS). In order to have good cell separation, we noted that the optimum amount of antibodies and incubation times differ for MEF and MCF7 cells. For each biological replicate we used duplicate 'Mix' and 'Co-culture' samples – one used to enrich for MCF7 cells and the other to enrich for MEFs. Cells were incubated on ice with magnetic bead-conjugated anti-CD326 microbeads (Miltenyi Biotec GmBH), as described in *Supplementary file 1 - table 1*.

Thereafter, an additional 1 ml of cold, complete DMEM (with FBS) was added to the cell suspension and sorted using MACS LS columns (Miltenyi Biotec GmBH). The columns were washed two times with 1 ml of cold DMEM each. The flow-through contains the CD326-negative MBS-MEF fraction, while the CD326-positive MCF7 fraction remains attached to the magnetic bead column. The MCF7 fraction was eluted with 3 ml of cold DMEM using the supplied plunger. To increase the purity of the MCF7 or MEF fraction, the flow-through (MEFs) or eluate (MCF7s) was sorted again using a fresh MACS LS column, as described above. A small aliquot (~1/10th) of the sorted MBS-MEF and MCF7 cell populations was then counterstained with Alexa-488-labeled anti-human CD326 (Biolegend; Cat #: 324210) and PE-labeled anti-mouse CD321 antibody (BD Biosciences, Cat #: 564908). For both antibodies, 5 µl of antibody per million cells were used and the sorting efficiency was checked using a flow cytometer (Attune NXT, Thermo Scientific). Samples belonging to replicate experiments were collected at the same time to reduce the batch effect. To evaluate the contamination in MEF-enriched samples, only the CD326-high and CD321-low quadrant was considered as contaminating MCF7 cells in MEF enriched fractions. To evaluate the level of MEF contamination in the MCF7-enriched sample, the CD326-low and CD321-high quadrant was considered as contaminant. These numbers are summarized in *Supplementary file 1 - table 1*. Note that there is a low percentage of double CD321-CD326 positive-stained cells (upper right quadrant) which is probably due to non-specific staining with the antibodies, as it appears in single cultures as well (*Figure 1—figure supplement 1*, *Supplementary file 1 - table 1* '% False positive'). There is also a small percentage of unstained cells (lower left quadrant). These are probably MEFs, since these are relatively abundant in MEF-enriched populations (0.1–2.8%), but rare events in the MCF7-enriched populations (0–0.08%).

## RNA extraction, quality control, library preparation, and sequencing

Total RNA was extracted using the NucleoSpin RNA Mini kit (Macharey Nagel) and RNA integrity checked using an Agilent Tapestation 2100, while the RNA concentration was measured using a Nano-Drop microvolume spectrophotometer and Qubit 2.0 BR assay. Transfer of human β-actin from MCF7 to MBS-MEF was verified by RT-PCR and RT-qPCR using oligos specific for human β-actin (*Supplementary file 2* – PCR primers). Poly-A(+) RNA sequencing libraries were prepared using the NEBNext Ultra II Directional RNA Library Prep Kit (New England Biolabs). Two µg of RNA per sample were processed in two separate reactions (separate technical replicas) to increase depth. Libraries were amplified by 8 PCR cycles and sequenced on a Novaseq 6000 SP1 flowcell using 2x150 bp paired-end reads. The processed files have been deposited in the Gene Expression Omnibus (GEO) Database under the accession number GSE185002 (https://www.ncbi.nlm.nih.gov/geo/query/acc.cgi?acc=GSE185002).

## Post processing and alignment to reference genomes

Identification of human-specific reads was done in multiple sequential steps (Figure S1B). First, *fastq* reads were trimmed from their adapter using *cutadapt* and aligned to the human genome (hg38) using STAR (v 2.7.3 a) (*Dobin and Gingeras, 2016*; *Martin, 2011*). The alignment was done without soft clipping (*--alignEndsType EndToEnd*) and using *--outFilterMismatchNoverLmax 0.0*. Second, the alignment was filtered to contain only paired-end reads with unique alignment. Third, uniquely-aligned reads were re-aligned to the mouse genome (mm10) using the same criteria. Finally, for every read the number of differences to the human genome was compared to the number of differences to the mouse genome. Only the reads that aligned to the human genome with zero mismatches and aligned to mouse with >2 differences were considered as *bonafide* human reads and were considered for further analysis. The same approach was applied to identify mouse-specific genes in human-enriched samples. HTseq was used to count the number of reads per gene. The read count was normalized to the library size of each sample (reads per million, RPM). All fastq data and raw read count data

have been deposited in the Gene Expression Omnibus (GEO) Database under the accession number GSE185002.

## Statistical analysis to detect transferred genes and differential gene expression

Exploratory analysis (hierarchical clustering and principal component analysis) of the count data was done using the base functions in R. To detect the most abundantly transferred human mRNAs we compared the normalized read count of the Co-culture samples to that of the Mix samples using an unpaired, one-tailed t-test, followed by multiple testing correction (Benjamini, Krieger, & Yekutieli procedure). Genes with a FC $\geq 2$ and False Discovery Rate (FDR)<5% were considered as enriched in the co-culture samples. Other statistical packages, such as DESeq2 or edgeR, which average the dispersion of the same gene across different samples are not useful here as we compared samples of different RNA content. Indeed, DESeq2 did not detect any significantly transferred genes between the Mix and Co-culture samples. We also performed a non-parametric rank analysis to check if the relative ranks of certain genes are different between Co-culture and Mix. Using this test, we failed to detect the transfer of β-actin as a significant observation (not shown), in contrast to our previous FISH results (*Dasgupta and Gerst, 2020*).

To estimate the proportion of the donor mRNAs that undergoes transfer, we defined the per gene 'percentage transfer' as: Percentage transfer = [[Avg. RPM (Co-culture) – Avg. RPM (Mix) x 100] / [Avg. RPM (MCF7 Single Culture)]]. To minimize the noise from the low-expressed genes, genes with RPM <100 for the Single culture and <10 for the Co-culture and FC <2 were not considered.

To estimate the gene expression level of the native mouse transcriptome, and to test the differential expression between the different conditions, the HTseq counts were processed using DEseq2 for normalization and differential expression testing (*Love et al., 2014*). A gene was considered as having a differential expression if the absolute value of the log2 fold change between conditions was at least 1, the $p_{adj}$ (false discovery rate; FDR)<0.05 (Benjamini-Hochberg correction), and the gene has at least 50 normalized counts in at least two samples.

## Motif enrichment analysis of transferred genes

The top 283 transferred genes (as obtained in the previous section) were converted to their ENSEMBL transcript ID of the longest splice variant and the 5' UTR, CDS and 3' UTR of each gene extracted using the 'table browser' tool of the UCSC genome browser by annotating to the GENCODE V38 database (exported in FASTA format). The 5' UTR, CDS and the 3' UTR of the genes were then analyzed separately using the "Motif Discovery" mode of MEME Suite using the following inputs: Mode: Classic; Input Type: DNA/RNA; Zero or one occurrence per sequence (zoops); maximum width: 15; return top 3 motifs. No motifs were identified among the two UTRs, while the top three motifs in the coding sequence are shown. As a comparison, similar analysis were done on groups of 280–300 genes among the pool of less transferred genes and the top three motifs of each group were identified. To scan for the presence of previously reported EV-targeting RNA motifs in the set of highly transferred RNAs, FIMO software (Find Individual Motif Occurrences) was used with default parameters (*Grant et al., 2011*). Three sets of least transferred RNAs, as used to discover transfer-promoting motifs by MEME analysis, were used as controls. Threshold p-value was determined empirically to discard non-specific matches. The analysis was performed on full length RNAs (i.e. 5' UTR, CDS, 3' UTR).

## Doxycycline induction experiments

In a 12-well plate, inducible tdTomato cells ($2.5 \times 10^4$ for co-culture and $5 \times 10^4$ for monoculture) were plated onto fibronectin-coated glass cover slips in the presence of 4 µg/ml, 2 µg/ml, or 0 µg/ml Doxycycline. After 24 hr, the medium and Doxycycline were replaced, and WT MEFs ($4 \times 10^4$ for co-culture and $8 \times 10^4$ for monoculture) were plated on top of the tdTomato cells. In parallel, a WT MEF monoculture sample (negative control) was cultured in the presence of 4 µg/ml of Doxycycline. After 16 hr, all samples were subjected to smFISH.

## Single molecule fluorescent in situ hybridization (smFISH)

smFISH was carried out according to a previously described protocol (*Haimovich et al., 2017*; *Haimovich and Gerst, 2018*; *Haimovich and Gerst, 2019*). A set of Cy5-labeled FISH probes to

detect tdTomato mRNA and Quasar (Q) 570-labeled FISH probes for KRT8 mRNA (*Zuckerman et al., 2020*) were generous gifts from Shalev Itzkovitch and Igor Ulitsky (Weizmann Institute of Science), respectively. Q570-labeled FISH probes for CCND1 and PSAP were obtained from LGC Biosearch Technologies. Cy5-labeled probes for MBS were reported previously (*Dasgupta and Gerst, 2020*; *Haimovich et al., 2017*). To stain actin filaments, slides were incubated with 25 nM Phalloidin-FITC (Sigma-Aldrich) during the first post-hybridization wash. The sequences of the labeled nucleotides are provided in *Supplementary file 2* – FISH probes.

## Widefield imaging

Images of tdTomato smFISH experiments and live imaging of GFP-ps and TagRFP-T-ps-expressing cells were captured using a Zeiss AxioObserver Z1 DuoLink dual camera imaging system equipped with an Illuminator HXP 120 V light source, Plan Apochromat 100×1.4 NA oil-immersion objective, and a Hamamatsu Flash 4 sCMOS camera. Thirty 0.2 μm step z-stack images were taken for smFISH and nine 0.5 μm step z-stack images were taken for the live imaging, using a motorized XYZ scanning stage 130×100 Piezo, and ZEN2 software at 0.0645 μm per pixel or 0.130 μm per pixel. All other images were captured with a Nikon Ti2 series inverted microscope equipped with a Du-888 CCD camera and 100 x oil-immersion objective. Thirty 0.2 μm step z-stack images were taken using NIS Elements software at 0.130 μm per pixel.

## Image analysis and presentation

The number of mRNAs (FISH spots) were scored using a MATLAB based GUI program, FISH-Quant (FQ), as described (*Mueller et al., 2013*; *Tsanov et al., 2016*). Briefly, outlines of both the cell and nucleus for all cells were demarcated by the 'Define Outlines' functions. In co-culture samples, donor cells were easily distinguished from acceptor cells due to the great number of endogenously expressed query RNA spots in the cytoplasm, as well as the oft presence of transcription sites in the nucleus. In addition, nuclear DAPI labeling was also used as a means to differentiate cells in human-mouse co-cultures: the human nucleus appears less granular and more elongated, whereas the murine nucleus appears granular and more rounded/ovular (see *Figure 4—figure supplement 1*). tdTomato donor cells were also distinguishable from acceptor cells due to their high level of tdTomato protein fluorescence. Thereafter, FISH spots were characterized from the donor cells (as the positive control) and detection settings were then applied to all cells in the same batch of analysis. Typically, one donor cell is filtered for background removal (typical parameters for filtering in FQ were Kernel BGD XY, Z=6, 5; Kernel SNR XY, Z=0.5, 1) and the individual spots were approximately determined by fitting them to a 3D Gaussian function. Due to high background in the nucleus, spots were only detected in the cytoplasm. Apparent non-specific spots were discarded by adjusting for the intensity and the width of Gaussian function. The detection settings thus obtained were applied to all cells (e.g. other donor cells, acceptor cells only, and the acceptor cells in co-culture) using the 'batch processing' tool. Once FQ preliminarily counted the number of spots in all the cells, the image parameters (e.g. Sigma XY, Sigma Z, Signal Amplitude and Background) were adjusted so as to have as minimum number of spots as possible in cells that do not express the relevant mRNA (i.e. acceptor cells alone). The modified detection settings were then re-applied to all cells to determine the number of spots in each cell. For image presentation, representative images from each condition are presented as maximum projection and were minimally adjusted for brightness and contrast in Fiji (*Schindelin et al., 2012*) and a RGB picture was generated by assigning an appropriate Look Up Table (LUT) to each channel. For TNT in FISH images, brightness was increased so the cell body appears over-exposed, to allow visualization of the thin TNTs. We used FIJI's bandpass filter (large structures down to 40 pixels, small structures up to 3 pixels, tolerance of direction 5%) for improved representation of the TNTs.

## Quadrapod and conditioned media experiments

To check for mRNA transfer from MCF7 cells to MBS-MEFs via EVs, MBS-MEFs and MCF7 cells were either spatially separated (i.e. the quadrapod experiment) or MBS-MEFs were cultured in conditioned media derived from MCF7 cells. For the quadrapod experiment, 15 cm cell culture plates were cut to 12 cm wide circular shapes from 15 cm tissue culture plates at the Scientific Instrumentation work-shop of the Weizmann Institute. Four Perspex legs (~2 mm in height) per disk were set in a square arrangement, as indicated in *Figure 6—figure supplement 1*. Disks were perforated with 1 mm-wide

holes to allow proper aeration and avoid formation of bubbles. All disks were washed in 70% EtOH for minimum of 24 hr before use. Disks were stored under sterile conditions at room temperature until used. First, $3x10^6$ MCF7 cells were cultured on fibronectin-coated disks for 6 hr. Second, $3x10^6$ MBS-MEF cells were seeded onto a separate 12 cm fibronectin-coated disk and cultured in a 15 cm plate. After 10 min of MBS-MEF cells attaching to the fibronectin-coated disk, the disk containing MCF7 cells was then placed over the disk containing MBS-MEF cells, so that the two cell types face each other, and were cultured for an additional 14 hr. For the conditioned media experiment, culture media was harvested from a 14 hr culture of $3x10^6$ MCF7 cells, centrifuged for 10 min at 500 x *g* and added to $3x10^6$ MBS-MEFs in a separate culture plate. Empty quadrapods were used in Single MEF, Co-culture and conditioned media samples to equalize any effect of aeration. Cells were then trypsinized and sorted using magnetic beads as described above, prior to the detection of human mRNAs in the purified MEF-enriched fraction by qRT-PCR. qPCR primer sequences are found in *Supplementary file 2*. Note that the hActb primers for RT-qPCR are different from the primers used for semi-quantitative PCR (*Figure 1C*) due to a non-specific band seen with these primers (*Figure 1—source data 1*).

## Statistical analysis

The results from *Figures 4 and 6*, and *Figure 5—figure supplement 1* were compared by using the one way ANOVA followed by a pair-wise Tukey's test to estimate the corrected P value for each pair of indicated corrections. Values from *Figure 5—figure supplement 1* are tabulated in *Supplementary file 1 - table 2*. All indicated calculations were performed using GraphPad Prism software Version 7 (GraphPad Software, Inc).

## Acknowledgements

We thank Ester Feldmesser and Igor Ulitsky (Weizmann Institute of Science, Israel) for assistance with the bioinformatics analysis; Michael Wigler and Siran Li (Cold Spring Harbor Laboratories, NY, USA) for generously performing scBAG-seq experiments to detect mRNA transfer; and Shalev Itzkovitz (Weizmann Institute of Science, Israel) for the gift of reagents. This work was funded by grants to JEG from the Joel and Mady Dukler Fund for Cancer Research, the Jean-Jacques Brunschwig Fund for the Molecular Genetics of Cancer, a Proof-of-Principle Grant from the Moross Integrated Cancer Center, the Kekst Family Institute for Medical Genetics (Weizmann Institute of Science), the German-Israel Foundation (GIF; I-1461–412.13/2018) (JEG and ML), and the US-Israel Binational Science Foundation-National Science Foundation (#2015846) (JEG and RHS).

## Additional information

### Competing interests

Robert H Singer: Reviewing editor, *eLife*. The other authors declare that no competing interests exist.

### Funding

| Funder | Grant reference number | Author |
|---|---|---|
| German-Israeli Foundation for Scientific Research and Development | I-1461-412.13/2018 | Markus Landthaler Jeffrey E Gerst |
| US-Israel Binational Science Foundation- National Science Foundation | 2015846 | Robert H Singer Jeffrey E Gerst |
| Joel and Mady Dukler Fund for Cancer Research | | Jeffrey E Gerst |
| Jean-Jacques Brunschwig Fund for the Molecular Genetics of Cancer | | Jeffrey E Gerst |

| Funder | Grant reference number | Author |
|---|---|---|
| Moross Integrated Cancer Center | | Jeffrey E Gerst |
| Kekst Family Institute for Medical Genetics | | Jeffrey E Gerst |

The funders had no role in study design, data collection and interpretation, or the decision to submit the work for publication.

## Author contributions

Sandipan Dasgupta, Data curation, Formal analysis, Validation, Investigation, Visualization, Methodology, Writing - original draft; Daniella Y Dayagi, Data curation, Formal analysis, Investigation, Visualization; Gal Haimovich, Conceptualization, Data curation, Formal analysis, Supervision, Validation, Investigation, Visualization, Methodology, Writing - original draft, Writing - review and editing; Emanuel Wyler, Data curation; Tsviya Olender, Data curation, Software, Formal analysis, Methodology, Writing - original draft; Robert H Singer, Markus Landthaler, Resources, Funding acquisition; Jeffrey E Gerst, Conceptualization, Resources, Supervision, Funding acquisition, Visualization, Writing - original draft, Project administration, Writing - review and editing

## Author ORCIDs

Daniella Y Dayagi http://orcid.org/0000-0003-2310-2416
Gal Haimovich http://orcid.org/0000-0002-3360-5108
Emanuel Wyler http://orcid.org/0000-0002-9884-1806
Robert H Singer http://orcid.org/0000-0002-6725-0093
Jeffrey E Gerst http://orcid.org/0000-0002-8411-6881

## Decision letter and Author response

Decision letter https://doi.org/10.7554/eLife.83584.sa1
Author response https://doi.org/10.7554/eLife.83584.sa2

# Additional files

## Supplementary files

• Supplementary file 1. This file contains Tables 1-3. Table 1. The table contains the conditions of cell sorting and percentage of cell populations after antigen-based magnetic bead cell sorting. Table 2. Results from post-hoc Tukey Test following one way ANOVA from *Figure 5—figure supplement 1C* – Detection of tdTomato mRNA transfer from donor MBS-MEFs expressing tdTomato-β-actin fusions to WT MEFs. Table 3. Number of occurrences of EV-mRNA targeting motifs in transferred RNAs of the same RNA sets used in *Figure 5—figure supplement 2*.

• Supplementary file 2. This file contains lists of FISH probes and PCR primers sequences.

• MDAR checklist

• Source data 1. This file contains 10 supplementary tables.

## Data availability

The processed RNA sequencing files have been deposited in the Gene Expression Omnibus (GEO) Database under the accession number GSE185002. All materials generated are available upon request.

The following dataset was generated:

| Author(s) | Year | Dataset title | Dataset URL | Database and Identifier |
|---|---|---|---|---|
| Dasgupta S, Dayagi D, Olender T, Haimovich G, Wyler E, Singer R, Landthaler M, Gerst JE | 2021 | NGS Analysis of human mRNAs in mouse cells following a 12-hr 2D-coculture | https://www.ncbi.nlm.nih.gov/geo/query/acc.cgi?acc=GSE185002 | NCBI Gene Expression Omnibus, GSE185002 |

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
