## [Editor Report]

This study presents an important finding on the characterization of cell contact-dependent transfer of mRNAs between human MCF7 breast cancer cell line, and immortalized mouse embryo fibroblasts (MEFs) grown in co-culture. The evidence supporting the conclusions is compelling, with rigorous data analysis and multiple approaches to address the specific questions of the sequences of the transferred mRNAs, the presence of specific sequences targeting this transfer, the cell-contact-dependent mechanisms, and the transcriptional consequences of the transfer. This work will be of interest to cell biologists and biologists.

---

## [Decision Letter]

**Decision letter after peer review:**

Thank you for submitting your article "Global analysis of contact-dependent human-to-mouse intercellular mRNA and lncRNA transfer in cell culture" for consideration by *eLife*. Your article has been reviewed by 2 peer reviewers, and the evaluation has been overseen by a Reviewing Editor and Vivek Malhotra as the Senior Editor. The following individual involved in the review of your submission has agreed to reveal their identity: Christel Brou (Reviewer #1).

In this paper, the authors show that tunneling nanotubes or TNTs are used by cells to transfer full-length mRNAs. The data show that as much as 1% of the endogenous mRNA are passed between cells by this procedure. The transferred mRNA affect the transcriptome of the acceptor cells thus highlighting the significance of this nanotube-mediated trafficking of mRNA between cells.

Essential revisions:

1) The identification of transferred RNAs relies heavily on the purity of the isolated cell populations and while the reviewers recognized that the authors did a good job, the isolated MEF samples may have uncharacterized components of MCF7 cells. Given the low level of detectable transferred RNAs it's really crucial to clarify the confidence of the measurements, with also more replicates. The purity of the sorted populations could be strengthened by qPCR-based assessments as suggested by one of the reviewers.

2) The authors should tone down the generality of the findings, as the results here on cell-to-cell contact in co-culture cells for 5 RNAs may not be valid transcriptome-wide. Furthermore some statements about the potential significance of these changes to cancer/immunity may be speculative.

3) Aspects of image acquisition and TNT characterization need to be improved to more convincingly correlate cell contact-dependent transfer of mRNA and transfer through TNTs. Also, two cell types beyond nuclear morphology need to be better identified.

*Reviewer #1 (Recommendations for the authors):*

The conclusions of this paper are mostly well supported by data, but some aspects of image acquisition and TNT characterization need to be improved to more convincingly correlate cell contact-dependent transfer of mRNA and transfer through TNTs (Figure 6). TNT should be better characterized as being membrane protrusions supported by actin and running above the substratum. Also, it would be better to clearly identify the two cell types beyond nuclear morphology that is not visible in this figure.

Regarding the differential changes in mouse transcriptome after coculture with MCF7 cells, the authors should better correlate these transcriptional changes to mRNA transfer and show that they do not result from other events (signaling pathways, transfer of organelles…). Are the transferred mRNAs translated, are the transcriptomes of receiving cells affected if the cocultures are performed in the presence of translation inhibitors?

*Reviewer #2 (Recommendations for the authors):*

The purity of the sorted populations could be further strengthened by a quantitative PCR-based assessment to determine the relative copy number of human versus mouse nuclear DNA in the isolated MEFs, as well as by expressing a fluorescent protein within MCF7 cells and assessing the presence of fluorescent cells by flow cytometry of the isolated MEF preparations.

It could also be informative to analyze the sorted populations by forward scatter as a measure of relative cell size. This might help to identify to what extent clusters of cells or cell fragments are present (especially as it relates to the 'double stained' and 'unstained' cells observed in the lower left and upper right quadrants of the profiles shown in Figure 1B and S1A).

As stated above, increasing the number of replicates might be helpful in revealing specific patterns of transferred RNAs.

Figures 4B, 4C, 5B, 5C, and S5C seem to present quantifications of multiple cells from a single experiment. The inclusion of at least another replicate would be useful.

The plots in Figures 2D, 2E, 2F, 2G, 3B, and 3C highlight specific RNAs by name, but the highlighted sets are only partly overlapping between plots, with some RNAs being pointed out in some plots but not others. It would be useful to either highlight the same RNAs in all plots or explain the rationale behind the current selections.

It seems to me that the results of Figures 6 and S8 indicate that mRNA transfer could at least partly occur via TNTs. The statements "we conclude that mRNA transfer in our co-culture system occurs predominantly via TNTs" (lines 426-427) and "we now provide strong evidence that transfer is via TNTs" (lines 529-529) sound overstated. For the same reason, I would suggest rewording the last sentence of the Abstract and Introduction.

The n value for the experiment of Figure 1D is not indicated.

---

## [Author Response]

Essential revisions:1) The identification of transferred RNAs relies heavily on the purity of the isolated cell populations and while the reviewers recognized that the authors did a good job, the isolated MEF samples may have uncharacterized components of MCF7 cells. Given the low level of detectable transferred RNAs it's really crucial to clarify the confidence of the measurements, with also more replicates. The purity of the sorted populations could be strengthened by qPCR-based assessments as suggested by one of the reviewers.

We invested much effort to determine the purity of the samples and to validate that our results do not stem from background contamination. We provide detailed answers to the specific reviewers’ concerns below. Note that the level of RNA detection in the RNA-seq samples was not low, rather it is that the level (percentage) of RNA transfer from donor cells to acceptor cells is low relative to expression in the donor cells.

2) The authors should tone down the generality of the findings, as the results here on cell-to-cell contact in co-culture cells for 5 RNAs may not be valid transcriptome-wide. Furthermore some statements about the potential significance of these changes to cancer/immunity may be speculative.

Per the reviewers’ comments we have toned down some of our statements (see below). However, we would like to emphasize that the results of cell-cell transfer in this and our previous publication did test a wide variety of mRNAs from different families and having various sub-cellular localizations. This indicates that the pathway we uncovered is more general than the few examples we present in our study. In addition, we now cite in the Discussion (lines 609-620 in the revised manuscript) a new pre-print recently posted to *bioRxiv* that show similar results of mRNA transfer in a human-mouse cell co-culture model. In relation to the statement about the significance of our findings to cancer/immunity, we actually used the term “speculate” in the text (lines 429, 577 in the original manuscript; lines 458, 627 in the revised manuscript). Thus, it should be clear that these statements are speculative until further research is performed.

3) Aspects of image acquisition and TNT characterization need to be improved to more convincingly correlate cell contact-dependent transfer of mRNA and transfer through TNTs. Also, two cell types beyond nuclear morphology need to be better identified.

We have now performed additional experiments that better characterize the TNTs and provide the detailed answers below.

Reviewer #1 (Recommendations for the authors):The conclusions of this paper are mostly well supported by data, but some aspects of image acquisition and TNT characterization need to be improved to more convincingly correlate cell contact-dependent transfer of mRNA and transfer through TNTs (Figure 6). TNT should be better characterized as being membrane protrusions supported by actin and running above the substratum.

We have now improved resolution of the TNTs and transferred RNAs observed therein in new smFISH experiments utilizing phalloidin staining. These new experiments (new Figure 6 and Videos 1 and 2) show actin-based TNTs with labeled transferred mRNAs residing therein and connecting cells, as well as extending over the substratum. Figure 6 from the original manuscript is now labeled as Figure 6—figure supplement 3.

Also, it would be better to clearly identify the two cell types beyond nuclear morphology that is not visible in this figure.

While nuclear morphology is helpful to distinguish between the two cell types in co-culture, the clearest and most useful way to distinguish between donor and acceptor cells is based on the expression level of the detected mRNA (*i.e.* as the RNA level in donor cells is much higher than in the acceptor cells, as seen in Figures 4B, 5A, 6, S3 and S5), as well as the clear presence of transcription sites in the donor cells (*e.g*. Figures 4B [KRT8], and 5A). In the case of experiments performed with tdTomato RNA, donor cells could be clearly identified by tdTomato protein fluorescence. The delineation between donor and acceptor cells used in the smFISH experiments was described in the Methods (Lines 790-797 in the original manuscript; lines 844-851 in the revised manuscript).

Regarding the differential changes in mouse transcriptome after coculture with MCF7 cells, the authors should better correlate these transcriptional changes to mRNA transfer and show that they do not result from other events (signaling pathways, transfer of organelles…).

We note that we did not claim that the changes in endogenous mRNA expression levels arise solely due to RNA transfer. Our only claim was that these changes arise upon conditions of cell co-culture and when TNT formation occurs. Future co-culture studies measuring transcriptome-wide RNA transfer using the quadrapod co-culture system (which prevents cell-cell contact under co-culture conditions) versus co-culture conditions that allow cell-cell contact could eventually be performed, yet are currently beyond the scope of this paper. To make a clear distinction between TNT-dependent and -independent effects upon the endogenous transcriptome also requires further mechanistic knowledge of TNT-mediated mRNA transfer as well as the contribution of signaling pathways to the transcriptional control of co-cultured cells.

Are the transferred mRNAs translated.

The translation of transferred RNAs is an important question currently under investigation in the lab, but is beyond the scope of this paper which aimed solely to document the transferome.

Are the transcriptomes of receiving cells affected if the cocultures are performed in the presence of translation inhibitors?

In our previous publication (Haimovich *et al.* 2017), we demonstrated that short-term (*e.g.* 3hrs) translation inhibition does not inhibit mRNA transfer, but we did not try it for longer periods and did not examine the phenotype of acceptor cells in the presence of translation inhibitors. We suspect that such an experiment will not be conclusive, since long-term (*e.g.* 12-24hr) translation inhibition will likely result in pleiotropic changes that can affect many cellular pathways and alter cell viability. In the future, we aim to inhibit transfer of a specific mRNA (*e.g*. by co-expressing an MS2 aptamer-tagged RNA and the MS2 coat protein in the donor cells; Haimovich *et al.*, 2017) or its ability to undergo translation (*e.g.* by eliminating the start codon or adding a riboswitch to control translation) in order to test the phenotypic consequence of the transfer of individual RNAs on acceptor cells. However, this is beyond the scope of this paper.

Reviewer #2 (Recommendations for the authors):The purity of the sorted populations could be further strengthened by a quantitative PCR-based assessment to determine the relative copy number of human versus mouse nuclear DNA in the isolated MEFs, as well as by expressing a fluorescent protein within MCF7 cells and assessing the presence of fluorescent cells by flow cytometry of the isolated MEF preparations.

We used end-point DNA PCR in our previous publication (Dasgupta and Gerst, 2020) to show the purity of the sorted cells, although it was not done for the cells collected for this paper. In this paper we used semi-quantitative and quantitative PCR to show the purity of the populations after purification, and mitochondrial RNA as another measure of purity. Importantly, our co-culture samples are compared to the “Mix” control. Indeed, the RNA-seq data in the opposite direction (*i.e.* mouse-to-human transfer) showed the expected increase in Mix-derived samples vs the Co-culture-derived samples, in direct correlation with the small contaminations detected by the FACS analysis (lines 161-165 in the original manuscript, lines 167-171 in the revised manuscript). Thus, we believe that the efforts we invested in validating the purity of our separated populations are more than sufficient (but see also our response to the next comment).

It could also be informative to analyze the sorted populations by forward scatter as a measure of relative cell size. This might help to identify to what extent clusters of cells or cell fragments are present (especially as it relates to the 'double stained' and 'unstained' cells observed in the lower left and upper right quadrants of the profiles shown in Figure 1B and S1A).

As indicated above, it is unlikely that these populations contributed to the human mRNA signal in the MEFs, since the percentage of these populations was much higher in the Mix-derived samples than in the Co-culture-derived samples (lines 174-181 in the revised manuscript). Nevertheless, for due diligence, we took another look at the original FACS files of the isolated MEF populations. Looking at “large” particle sizes (*e.g.* 1-3% of all particles) showed these were exclusively single-stained for MEFs. The very small particles were distributed between single-stained for MEFs and unstained, with a minor population (~0.6% in Co-culture and ~1.6% in Mix and single MEF cultures) stained by the MCF7 antibody. So, it seems unlikely that any of these populations contributed to a “contamination” of the Co-culture-derived samples beyond that of the Mix-derived samples.

As stated above, increasing the number of replicates might be helpful in revealing specific patterns of transferred RNAs.

As stated above, unfortunately, we lack the resources to perform additional replicates of the RNA-seq experiments.

Figures 4B, 4C, 5B, 5C, and S5C seem to present quantifications of multiple cells from a single experiment. The inclusion of at least another replicate would be useful.

As mentioned above, these experiments were done in two replicates. This is now indicated in the figure legends.

The plots in Figures 2D, 2E, 2F, 2G, 3B, and 3C highlight specific RNAs by name, but the highlighted sets are only partly overlapping between plots, with some RNAs being pointed out in some plots but not others. It would be useful to either highlight the same RNAs in all plots or explain the rationale behind the current selections.

This is a good idea and we updated Figure 2E-G to show the same gene names. As for 2D, 3B and 3C: these plots show different properties and we highlighted a few example RNAs from the top and/or bottom of each plot. Therefore, we do not think highlighting the same RNAs is required in these cases. For interested researchers, the data is available in the supplementary excel files.

It seems to me that the results of Figures 6 and S8 indicate that mRNA transfer could at least partly occur via TNTs. The statements "we conclude that mRNA transfer in our co-culture system occurs predominantly via TNTs" (lines 426-427) and "we now provide strong evidence that transfer is via TNTs" (lines 529-529) sound overstated. For the same reason, I would suggest rewording the last sentence of the Abstract and Introduction.

In the revised paper we added more experiments that show the presence of mRNAs in *bona fide* TNTs (new Figure 6 and related movies). Clearly, we are unable to test each and every mRNA, but likely the simplest explanation is correct, *i.e.* that most if not all mRNAs transfer via TNTs. However, we added a few words to tone down our statements (lines 30, 112-113, 446, 547-548 in the revised manuscript), as requested.

The n value for the experiment of Figure 1D is not indicated.

We now added to the figure legend that this was done in triplicates.